# Unified Cross-Scale 3D Generation and Understanding via Autoregressive Modeling

## Abstract

3D structure modeling is essential across scales, enabling applications from fluid simulation and 3D reconstruction to protein folding and molecular docking. Yet, despite shared 3D spatial patterns, current approaches remain fragmented, with models narrowly specialized for specific domains and unable to generalize across tasks or scales. We propose Uni-3DAR, a unified autoregressive framework for cross-scale 3D generation and understanding. At its core is a coarse-to-fine tokenizer based on octree data structures, which compresses diverse 3D structures into compact 1D token sequences. We further propose a two-level subtree compression strategy, which reduces the octree token sequence by up to 8x. To address the challenge of dynamically varying token positions introduced by compression, we introduce a masked next-token prediction strategy that ensures accurate positional modeling, significantly boosting model performance. Extensive experiments across multiple 3D generation and understanding tasks, including small molecules, proteins, polymers, crystals, and macroscopic 3D objects, validate its effectiveness and versatility. Notably, Uni-3DAR surpasses previous state-of-the-art diffusion models by a substantial margin, achieving up to 256% relative improvement while delivering inference speeds up to 21.8x faster.

## 1 Introduction

3D structure modeling underpins a wide range of real-world applications, spanning the planetary-scale dynamics of celestial bodies to the angstrom-scale arrangements of atoms and electrons. At the macroscopic level, it enables 3D object reconstruction, computational fluid dynamics simulations, and climate forecasting; at the microscopic level, it supports protein structure prediction (Jumper et al., 2021), crystal generation (Jiao et al., 2023), molecular dynamics (Wang et al., 2018a), and molecular docking (Alcaide et al., 2024).

Despite these shared spatial principles, 3D modeling tasks have largely evolved in silos. Models tailored for macroscopic structures fail to transfer to microscopic domains, and even applications at similar scales rarely generalize. For instance, a model designed for crystal generation cannot be directly applied to protein folding (Xie et al., 2021; Jiao et al., 2023). This fragmented development hinders data reuse and results in redundant, highly specialized models rather than a unified solution.

To overcome this fragmentation, we propose Uni-3DAR, a unified autoregressive framework for cross-scale 3D generation and understanding. At its core is a tokenizer that efficiently compresses diverse 3D structures into discrete 1D token sequences. Leveraging these compressed sequences, our autoregressive model unifies generative and understanding tasks within a single architecture.

The proposed tokenizer uses an octree data structure to compress the full-size 3D grid both losslessly and efficiently. As illustrated in Fig. 2 (a) and (b), we construct an octree by recursively subdividing the space up to a maximum depth of $L$. To adapt to data sparsity, branches corresponding to empty regions are pruned, resulting in a maximum of $8^{L-1}$ leaf grid cells (but most will be pruned due to sparsity). We then introduce a fine-grained tokenization that encodes details within each occupied leaf cell (we call it a "3D patch"), such as atomic types and precise coordinates for molecules, or more general VQVAE tokens (Van Den Oord et al., 2017). Concatenating these tokens level by level produces a hierarchical, coarse-to-fine 1D token sequence that effectively represents the 3D structure (fig. 2(c)). Furthermore, we compress each two-level subtree (eight subcells) into a single 8-bit token instead of assigning an individual occupancy token to each node (Fig. 2 (d)). Since each

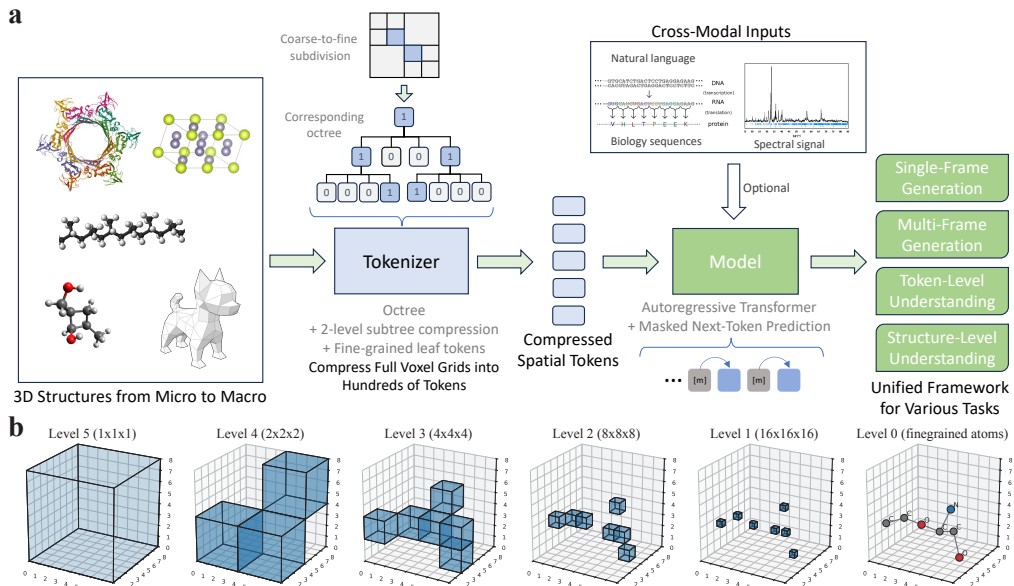

Figure 1: **Uni-3DAR Overview.** **(a)** A coarse-to-fine octree-based tokenizer converts 3D structures into 1D sequences (details in fig. 2). The tokens are modeled by an autoregressive transformer trained with masked next-token prediction (details in fig. 3) and can be optionally conditioned on cross-modal inputs (e.g., text, biological sequences, spectra). A single model supports single- and multi-frame generation as well as token- and structure-level understanding. **(b)** An example of octree from coarse level to fine level. Uni-3DAR generates tokens in a coarse-to-fine order: high-level occupancy tokens followed by level-0 tokens that capture local details (e.g., atom types and coordinates). The merits of octree over other 3D representations are discussed in Appendix A.

subcell is binary (empty or not), grouping eight subcells yields $2^8 = 256$ distinct states, reducing the sequence length approximately 8x and converting 8 binary classifications into one 256-class task.

However, the octree compression with empty tokens pruned disrupts the spatial mapping, meaning adjacent tokens no longer correspond to uniform intervals in the original 3D space. Unlike in 2D images with fixed patch positions, the model cannot reliably predict the next token without knowing its explicit target coordinates. We found that simply appending the next position to the current token yielded unsatisfactory results. To address this challenge, we propose a *masked next-token prediction* strategy. As illustrated in Fig. 3 (a), our method duplicates each token so that it appears twice with the same positional embedding. We then replace the first copy with a [MASK] token. The model still performs next-token prediction, but the prediction is made exclusively at the masked position. This setup ensures that the prediction is conditioned on the correct positional information of the intended target token, effectively resolving the issue of dynamic token positions. Although this approach doubles the sequence length, it achieves significant performance gains as validated in appendix D.2.

Uni-3DAR is built on several technical innovations: (1) a **coarse-to-fine octree-based tokenization** for efficient representation, (2) a **2-level subtree compression** to reduce sequence length, (3) a **unified fine-grained structural representation (for "3D patch")** to capture local details, and (4) a **masked next-token prediction** strategy to handle dynamic token positions, which enable our key contributions:

*1. Unified Cross-Scale 3D Modeling.* Leveraging the proposed coarse-to-fine tokenizer, Uni-3DAR can process a wide range of 3D structures, from macroscopic to microscopic scales.

*2. Unified Generation and Understanding.* Uni-3DAR seamlessly unifies 3D structural generation and understanding tasks within a single framework. As illustrated in Fig. 3 (b), different tasks use distinct tokens, ensuring clear separation without interference.

*3. High Efficiency.* Thanks to the octree and two-level subtree compression, Uni-3DAR represents 3D structures with far fewer tokens. For example, while O Pinheiro et al. (2023) requires $32^3 = 262,144$ tokens for a small molecule, Uni-3DAR needs only hundreds, and can scale to large proteins with thousands of atoms with deeper octree levels (section 3.4). Moreover, appendix D.3 shows that Uni-3DAR is approximately 21.8x faster than prior diffusion-based models.

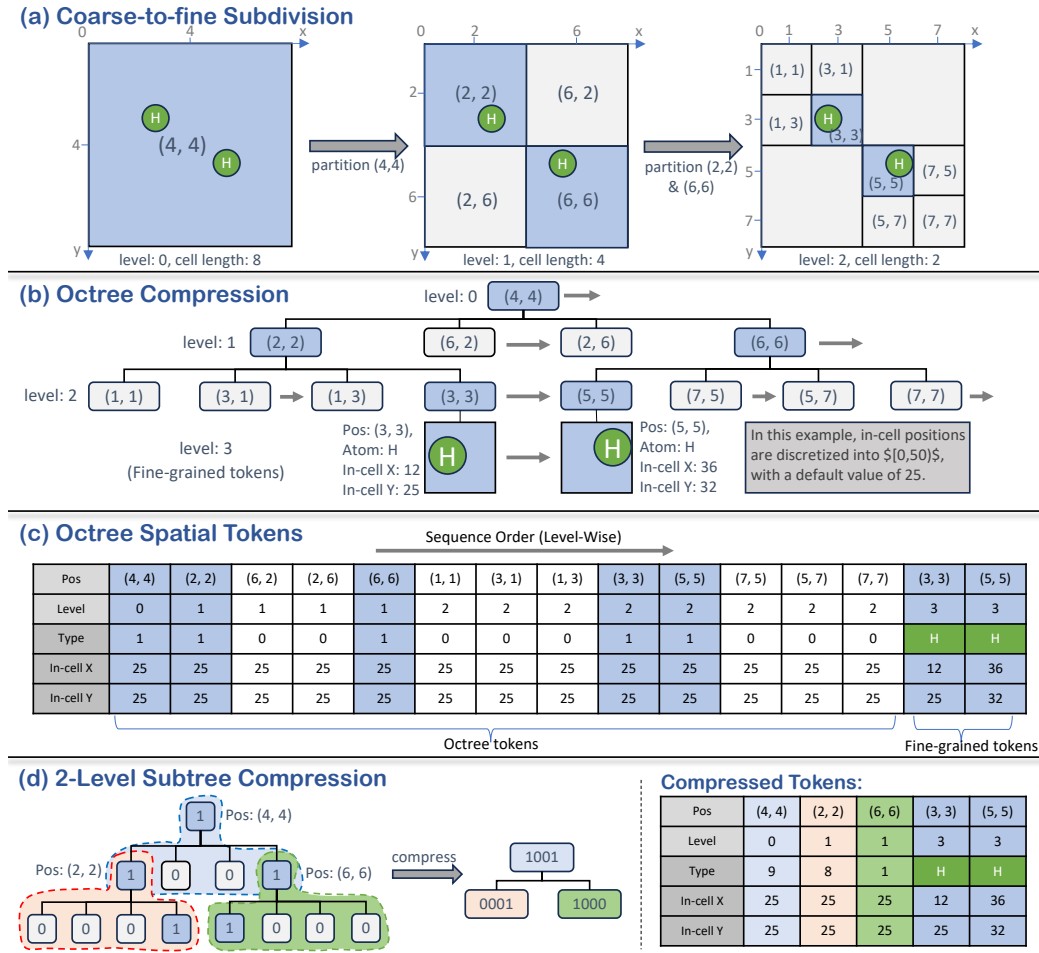

Figure 2: Overview of Uni-3DAR tokenization (illustrated in 2D using quadtree for clarity). **(a)** Adaptive coarse-to-fine subdivision of grid cells, where darker nodes indicate non-empty cells that can be further partitioned. **(b)** This partitioning process constructs an octree, providing a lossless compression of the full-size 3D grid. **(c)** Uni-3DAR's tokenization consists of two components: hierarchical spatial compression via an octree and fine-grained structural tokenization. Each node's position is determined by its tree level and cell center. **(d)** The proposed 2-level subtree compression reduces the octree tokens by 8x (4x in the illustrated quadtree).

*4. High Accuracy.* Extensive experiments across diverse tasks—including macroscopic 3D shape generation (table 4), molecular (table 1), crystal generation (table 2), protein pocket prediction (table 6), molecular docking (table 7), and molecular pretraining (tables 8 and 9)—demonstrate Uni-3DAR's superior or competitive performance compared to existing methods. Notably, Uni-3DAR consistently outperforms diffusion-based models. Ablation studies (table 12) highlight the benefits of unifying generation and understanding and validate the effectiveness of each component.

## 2 METHOD

### 2.1 DYNAMIC COARSE-TO-FINE TOKENIZATION FOR 3D STRUCTURES

3D structures are inherently sparse: at the microscopic scale, most space is empty except for scattered atoms; at the macroscopic level, detailed representations are only needed at object surfaces, with most volume remaining empty. Using a full-size voxel grid is thus highly inefficient. To address this, we propose a hierarchical, coarse-to-fine tokenization of 3D structures that exploits this sparsity. As shown in Fig. 1, our approach consists of two parts: (1) a hierarchical compression of 3D space using an octree, and (2) a fine-grained structural tokenization.

The first component is the octree, an efficient data structure for lossless 3D grid compression. Starting with a single cell covering the entire structure, we recursively subdivide it: if a cell contains atoms, it is partitioned further. Each subdivision halves each dimension, producing $2^3 = 8$ equal subcells (hence "octree"). This process continues for $L$ levels. If $c_0$ is the root cell length, the

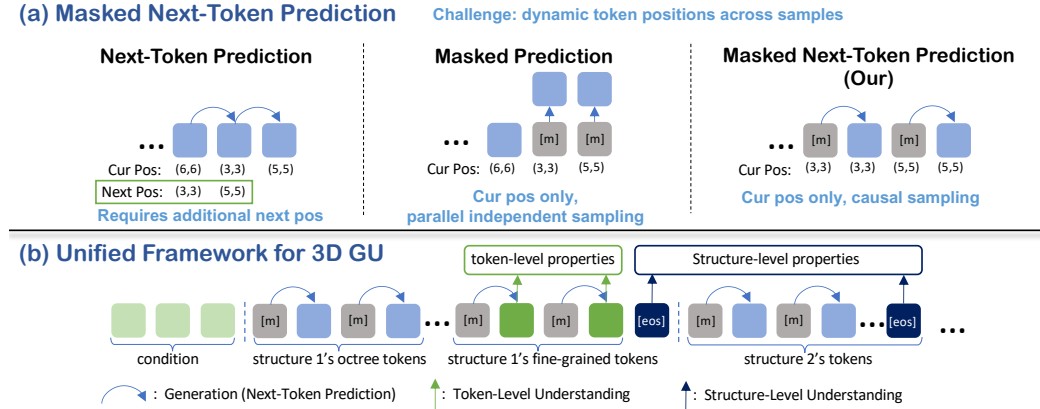

Figure 3: **(a) Masked Next-Token Prediction.** To handle the challenge of dynamically positioned tokens in sparse 3D structures, Uni-3DAR decouples position and content generation. Unlike standard next-token prediction, we first infer the next token's position from the octree hierarchy, place a "[MASK]" token, and then have the model predict only its content (e.g., occupancy or fine-grained properties). **(b) Unified Framework for 3D Generation and Understanding.** The Uni-3DAR architecture is a versatile, multi-task model. It supports autoregressive generation of complex 3D structures (blue arrows) and can be prompted to perform both token-level (green arrows) and structure-level (blue box) understanding tasks within a single framework.

cell length at level $L-1$ is $c_{L-1} = c_0/2^{L-1}$. We refer to these leaf subcells as fine-grained "3D patches," which are then tokenized as detailed in the following paragraph.

The second component is the fine-grained tokenization of structural details. While the octree effectively identifies coarse, non-empty regions, it lacks finer details such as atom types and precise coordinates (microscopic) or surface features (macroscopic). Although using deeper octrees can capture more detail (Ibing et al., 2023), this approach becomes inefficient due to the rapidly increasing number of tokens. Instead, inspired by the use of 2D image patches (Alexey, 2020), we treat the contents of each final-level non-empty region as a "3D patch." These patches can be processed in various ways; for instance, they can be quantized into discrete tokens for autoregressive prediction, similar to VQ-VAE (Van Den Oord et al., 2017), or modeled using a patch-level diffusion loss for continuous vector representations (Li et al., 2024b) (ablation in Table 12). In our experiments, we demonstrate this flexibility by using raw atom types and coordinates as fine-grained tokens for microscopic data (we set the patch size to ensure each 3D patch only contains one atom), and VQ-VAE tokens for macroscopic data. More details are in Appendix B.

Finally, we concatenate tokens level by level. Beyond token content, we represent each token's positional information using its tree level and the spatial coordinates of its cell center. For instance, the root cell is at level 0 with a center at $(c_0/2, c_0/2, c_0/2)$. During autoregressive prediction, since octree tokens are dynamically unfolded level by level, the positions of all tokens at the current level are known based on the predictions from previous level. This explicit knowledge of the token position is crucial, as autoregressive models predict only token content.

**2-Level Subtree Compression** Although octree tokenization avoids cubic cell growth, it remains inefficient for large 3D structures. Each level has up to $8N$ tokens ($N$ = non-empty final-level cells), totaling up to $8NL$ tokens across $L$ levels, about two orders of magnitude larger than $N$. To reduce this, we introduce 2-level subtree compression, merging a parent and its 8 children into a single token. As the parent's type is always 1, the subtree is fully represented by its 8 children's types, yielding $2^8 = 256$ possible states. This cuts token count by $8\times$, down to at most $N(L-1)$ tokens. For positional information of the compressed nodes, we retain the their parent's center and level.

## 2.2 MASKED NEXT-TOKEN PREDICTION FOR DYNAMIC TOKEN POSITIONS

In standard autoregressive models, such as those for text, token positions follow a fixed, sequential order (e.g., token $i + 1$ always follows token $i$). This static structure makes the next token's position implicitly known, obviating the need for its explicit prediction. In contrast, our coarse-to-fine 3D tokenization generates a token sequence where positions are *dynamic* and vary across different structures. This variability introduces a significant challenge: inferring the next token's position becomes non-trivial, making it preferable to provide this information to the model explicitly.

A straightforward approach is to encode both the current and next positions within each token (Ibing et al., 2023). However, we found this method leads to suboptimal performance (Table 12). We hypothesize that the unpredictable nature of the next token's position introduces noise that degrades the current token's representation. This intuition is supported by prior work like Yan et al. (2022), which decouples position and content prediction into two separate transformer modules.

Another promising direction is masked prediction (Chang et al., 2022; Li et al., 2024b), where a model predicts the content of a masked token given its position. This has proven effective for generative tasks with non-sequential or randomized token orders (Li et al., 2024b; Pang et al., 2024). However, directly applying conventional masked prediction to our framework is problematic. First, it typically relies on bi-directional attention, whereas our hierarchical tokenization unfolds uni-directionally. Second, it often requires parallel, non-causal sampling, which necessitates complex, rule-based inference strategies to balance performance and efficiency (Li et al., 2024b).

To resolve these issues, we introduce **Masked Next-Token Prediction (MNTP)**, a simple yet effective method that integrates masked prediction into a standard autoregressive framework. The core idea is to duplicate each token. For a given token, we first generate a placeholder with its content replaced by a special [MASK] symbol while retaining its correct position. This is immediately followed by a second token at the *same position* but with the true content. The model's objective is to predict the content of this second token, conditioned on the [MASK] token and all preceding tokens.

This formulation effectively reframes next-token prediction as a masked prediction task: the model is given a position with a mask and asked to fill in the content. This approach elegantly sidesteps the challenge of predicting dynamic next positions. Compared to conventional masked prediction, MNTP preserves a strictly causal, uni-directional attention flow, eliminating the need for complex sampling schemes. While this duplication doubles the sequence length, we demonstrate in Appendix D.2 that the substantial performance gains justify this trade-off. Furthermore, through targeted optimizations discussed in Appendix D.3, the impact on inference latency is modest, with only a 15%–30% increase compared to standard next-token prediction (appendix D.3).

## 2.3 Unified 3D Generation and Understanding Framework

By integrating techniques in sections 2.1 and 2.2, Uni-3DAR provides a unified framework for a wide range of 3D tasks (Figure 3(b)). The model architecture assigns distinct roles to different token types, enabling it to handle four primary task categories individually or jointly:

1. **Single-Frame Generation** (sections 3.1, 3.3, 3.4 and 3.6): Generating a single 3D structure, either unconditionally or conditioned on external modalities like text or chemical properties. This is accomplished using the masked tokens for autoregressive generation.

2. **Multi-Frame Generation** (sections 3.2 and 3.5) Autoregressively producing a sequence of 3D structures (multiple octrees), such as a molecular dynamics trajectory, molecular docking, or pocket-based generation. Each frame is distinguished by a unique frame-index embedding.

3. **Token-Level Understanding** (section 3.4) Predicting properties of local components (e.g., atomic forces or partial charges) by attaching a prediction head to the fine-grained tokens.

4. **Structure-Level Understanding** (section 3.6): Predicting global properties of an entire structure (e.g., solubility, toxicity) via a prediction head on the final "[EoS]" token. This allows Uni-3DAR to be pre-trained on large-scale unlabeled 3D data and efficiently fine-tuned for downstream tasks.

This versatile design allows for seamless joint training across these diverse tasks. Each token type serves a clear purpose: masked tokens drive generation, fine-grained tokens facilitate local understanding, and the "[EoS]" token enables global understanding.

Furthermore, the autoregressive nature of Uni-3DAR inherently supports multi-modal conditioning, which is critical for many scientific applications. For instance, a protein's amino acid sequence can guide the generation of its 3D fold. Similarly, experimental data like Powder X-ray Diffraction (PXRD) spectra can constrain the prediction of a crystal structure, a task we explore in section 3.2.

## 3 Experiments

We conducted extensive experiments to validate Uni-3DAR across diverse benchmarks. This section summarizes the key findings; complete implementation details settings can be found in appendices B and C. To ensure a fair comparison against existing methods, we trained separate model for each

Table 1: Performance comparison on unconditional 3D molecular generation. Results for UniGEM are marked with an asterisk (*) to indicate the use of additional molecular property information during training.

| | QM9 | | | | DRUG | |
| --- | --- | --- | --- | --- | --- | --- |
| | Atom Sta(%)↑ | Mol Sta(%)↑ | Valid(%)↑ | V × U(%)↑ | Atom Sta(%)↑ | Valid(%)↑ |
| Data | 99.0 | 95.2 | 97.7 | 97.7 | 86.5 | 99.9 |
| ENF (Garcia Satorras et al., 2021) | 85.0 | 4.9 | 40.2 | 39.4 | - | - |
| G-Schnet (Gebauer et al., 2022) | 95.7 | 68.1 | 85.5 | 80.3 | - | - |
| GDM (Hoogeboom et al., 2022) | 97.0 | 63.2 | - | - | 75.0 | 90.8 |
| GDM-AUG (Hoogeboom et al., 2022) | 97.6 | 71.6 | 90.4 | 89.5 | 77.7 | 91.8 |
| EDM (Hoogeboom et al., 2022) | 98.7 | 82.0 | 91.9 | 90.7 | 81.3 | 92.6 |
| EDM-Bridge (Wu et al., 2022) | 98.8 | 84.6 | 92.0 | 90.7 | 82.4 | 92.8 |
| GeoLDM (Xu et al., 2023b) | 98.9 | 89.4 | 93.8 | 92.7 | 84.4 | 99.3 |
| UniGEM* (Feng et al., 2024) | 99.0 | 89.8 | 95.0 | 93.2 | 85.1 | 98.4 |
| Uni-3DAR | **99.4** | **93.7** | **98.0** | **94.0** | **85.5** | **99.4** |

Table 2: Results on de novo crystal generation. Baseline results are taken from Xie et al. (2021).

| Data | Method | Validity (%) ↑ | | Coverage (%) ↑ | | Property ↓ | | |
| --- | --- | --- | --- | --- | --- | --- | --- | --- |
| | | Struc. | Comp. | COV-R | COV-P | $d_p$ | $d_E$ | $d_{elem}$ |
| Carbon-24 | FTCP (Ren et al., 2021) | 0.08 | – | 0.00 | 0.00 | 5.206 | 19.05 | – |
| | G-SchNet (Gebauer et al., 2019) | 99.94 | – | 0.00 | 0.00 | 0.9427 | 1.320 | – |
| | P-G-SchNet (Gebauer et al., 2019) | 48.39 | – | 0.00 | 0.00 | 1.533 | 134.7 | – |
| | CDVAE (Xie et al., 2021) | **100.0** | – | 99.80 | 83.08 | 0.1407 | 0.2850 | – |
| | DiffCSP (Jiao et al., 2023) | **100.0** | – | 99.90 | 97.27 | 0.0805 | 0.0820 | – |
| | Uni-3DAR | 99.99 | – | **100.0** | **98.16** | **0.0660** | **0.0289** | – |
| MP-20 | FTCP (Ren et al., 2021) | 1.55 | 48.37 | 4.72 | 0.09 | 23.71 | 160.9 | 0.7363 |
| | G-SchNet (Gebauer et al., 2019) | 99.65 | 75.96 | 38.33 | 99.57 | 3.034 | 42.09 | 0.6411 |
| | P-G-SchNet (Gebauer et al., 2019) | 77.51 | 76.40 | 41.93 | 99.74 | 4.04 | 2.448 | 0.6234 |
| | CDVAE (Xie et al., 2021) | **100.0** | 86.70 | 99.15 | 99.49 | 0.6875 | 0.2778 | 1.432 |
| | DiffCSP (Jiao et al., 2023) | **100.0** | 83.25 | **99.71** | 99.76 | 0.3502 | 0.1247 | 0.3398 |
| | FlowMM (Miller et al., 2024) | 96.85 | 83.19 | 99.49 | 99.58 | **0.239** | – | 0.083 |
| | Uni-3DAR | 99.89 | **90.31** | 99.62 | **99.83** | 0.4768 | **0.1237** | **0.0694** |

benchmark. We defer the investigation of joint training to future work. Uni-3DAR is robust to hyper-parameters, requiring no significant tuning and using a consistent setting across all tasks.

### 3.1 3D SMALL MOLECULE GENERATION

We assess Uni-3DAR on unconditional 3D molecular generation, a fundamental task challenged by the need to produce realistic conformations while accounting for molecular flexibility and diverse rotatable bonds. Our evaluation employs two standard benchmarks: **QM9** (Ramakrishnan et al., 2014b), a dataset of small molecules with up to 29 atoms, and **GEOM-DRUG** (Axelrod and Gomez-Bombarelli, 2022), which contains larger, more complex drug-like compounds with up to 181 atoms. Following the established protocols of Hoogeboom et al. (2022), we report on key metrics including Atom Stability, Molecule Stability, chemical validity (as determined by RDKit), and uniqueness. Bond types are inferred from the generated geometries to evaluate chemical correctness.

As shown in table 1, Uni-3DAR significantly outperforms all baseline models. On QM9, it achieves notable improvements in crucial metrics, reaching a Molecule Stability of 93.7% and a Validity of 98.0%, substantially exceeding the second-best methods. These results underscore Uni-3DAR's robust capability to generate high-quality, chemically valid molecules. Furthermore, Uni-3DAR surpasses UniGEM, a model that leverages additional molecular property information during training, using only 3D geometric data. This highlights the efficacy and robustness of our proposed model.

### 3.2 CRYSTAL GENERATION

We evaluate Uni-3DAR on crystal structure generation, a task distinct from organic molecules due to crystals' rigidity, symmetry, and periodicity. A crystal is represented by its lattice (parallelepiped unit cell) and atomic configurations. Uni-3DAR adopts a two-frame generation approach: first generating lattice vertices, then atom positions within the lattice. We consider three tasks: (1) de novo crystal generation (unconditional sampling), (2) crystal structure prediction (CSP) from given compositions, and (3) PXRD-guided CSP, which reconstructs crystal structures from PXRD signals and compositions, with practical relevance for real-world material discovery. For composition conditioning, we prepend a token from a multi-hot atom-type vector. PXRD data (0°–120° at 0.1° resolution) is converted into a 1200-dim vector, split into four segments, each as a conditional token—yielding

Table 3: Results on crystal structure prediction (CSP) and PXRD-guided CSP. For a fair comparison, we report UniGenX results obtained from the model trained from scratch, rather than using its default configuration that leverages large-scale datasets for additional pretraining and fine-tuning.

| Method | Carbon-24 | | MPTS-52 | | MP-20 | | MP-20 (PXRD-Guided) | |
|---|---|---|---|---|---|---|---|---|
| | Match Rate (%) ↑ | RMSE ↓ | Match Rate (%) ↑ | RMSE ↓ | Match Rate (%) ↑ | RMSE ↓ | Match Rate (%) ↑ | RMSE ↓ |
| CDVAE (Xie et al., 2021) | 17.09 | 0.2969 | 5.34 | 0.2106 | 33.90 | 0.1045 | – | – |
| DiffCSP (Jiao et al., 2023) | 17.54 | 0.2759 | 12.19 | 0.1786 | 51.49 | 0.0631 | – | – |
| FlowMM (Miller et al., 2024) | 23.47 | 0.4122 | 17.54 | 0.1726 | 61.39 | 0.0566 | – | – |
| UniGenX (Zhang et al., 2025) | 27.09 | 0.2264 | 29.09 | 0.1256 | 63.88 | 0.0598 | – | – |
| PXRDGEN (Li et al., 2024a) | – | – | – | – | – | – | 68.68 | 0.0707 |
| Uni-3DAR | **31.23** | **0.2194** | **32.44** | **0.0684** | **65.48** | **0.0317** | **75.08** | **0.0276** |

five tokens in total (one for composition, four for PXRD). Uni-3DAR's autoregressive framework integrates these tokens directly, avoiding extra encoders used in prior work (Li et al., 2024a; Lai et al., 2025). Following prior work (Xie et al., 2021; Jiao et al., 2023; Miller et al., 2024), we use Carbon-24 (Pickard, 2020), MP-20 (Jain et al., 2013), and MPTS-52 datasets. De novo generation is evaluated via validity, coverage, and property statistics (Xie et al., 2021), while CSP and PXRD-guided CSP are assessed by top-1 match rate and RMSE, using `StructureMatcher` (Ong et al., 2013) with the same thresholds as in (Miller et al., 2024).

Table 2 shows Uni-3DAR's performance on Carbon-24 and MP-20. On Carbon-24, Uni-3DAR outperforms baselines, especially in coverage, generating diverse and realistic structures. On MP-20, it achieves higher component validity while maintaining competitive results overall, highlighting its strength in producing chemically valid crystals. Table 3 summarizes CSP results across all datasets. Uni-3DAR consistently outperforms baselines, improving match rate by 4.14% on Carbon-24 and reducing RMSE from 0.0566 to 0.0317 on MP-20 (178% relative gain). On MPTS-52, it achieves 0.0684 RMSE, a 184% improvement despite higher complexity, demonstrating strong precision and generalization. For PXRD-guided CSP, Uni-3DAR surpasses PXRDGEN (Li et al., 2024a), raising the match rate from 68.68% to 75.08% and cutting RMSE from 0.0707 to 0.0276 (256% relative gain), showing exceptional accuracy in reconstructing crystals from PXRD data.

## 3.3 MACROSCOPIC 3D OBJECT GENERATION

To demonstrate its versatility beyond microscopic domains, Uni-3DAR was also evaluated on unconditional macroscopic 3D object generation, a fundamental task in 3D computer vision. We utilized three ShapeNet categories (*airplane, chair, car*) (Chang et al., 2015), where objects are represented as point clouds, and assessed using 1-NNA (with both Chamfer distance (CD) and earth mover distance (EMD) as our main metric following Vahdat et al. (2022). A distinctive aspect for this

Table 4: Unconditional 3D object generation results (1-NNA↓) on ShapeNet. The **best** and second-best results among the baselines are highlighted.

| Method | Airplane | | Chair | | Car | |
|---|---|---|---|---|---|---|
| | CD ↓ | EMD ↓ | CD ↓ | EMD ↓ | CD ↓ | EMD ↓ |
| r-GAN (Achlioptas et al., 2018) | 98.40 | 96.79 | 83.69 | 99.70 | 94.46 | 99.01 |
| l-GAN (CD) (Achlioptas et al., 2018) | 87.30 | 93.95 | 68.58 | 83.84 | 66.49 | 88.78 |
| l-GAN (EMD) (Achlioptas et al., 2018) | 89.49 | 76.91 | 71.90 | 64.65 | 71.16 | 66.19 |
| PointFlow (Yang et al., 2019) | 75.68 | 70.74 | 62.84 | 60.57 | 58.10 | 56.25 |
| SoftFlow (Kim et al., 2020) | 76.05 | 65.80 | 59.21 | 60.05 | 64.77 | 60.09 |
| SetVAE (Kim et al., 2021) | 76.54 | 67.65 | 58.84 | 60.57 | 59.94 | 59.94 |
| DPF-Net (Klokov et al., 2020) | 75.18 | 65.55 | 62.00 | 58.53 | 62.35 | 54.48 |
| DPM (Luo and Hu, 2021) | 76.42 | 86.91 | 60.05 | 74.77 | 68.89 | 79.97 |
| PVD (Zhou et al., 2021) | 73.82 | 64.81 | 56.26 | 53.32 | 54.55 | 53.83 |
| LION (Vahdat et al., 2022) | 67.41 | 61.23 | 53.70 | 52.34 | 53.41 | 51.14 |
| Uni-3DAR (Ours) | **67.35** | **61.09** | **53.11** | **51.78** | **53.35** | **50.89** |

task is Uni-3DAR's processing of an input object as $512 \times 512 \times 512$ voxels, and the resulting 3D patches (fine-grained structural tokens) are defined as $16 \times 16 \times 16$ voxels. Each patch is quantized using VQVAE. As shown in Table 4, Uni-3DAR exhibits highly competitive, often superior, performance against established baselines (Yang et al., 2019). More details are in Appendix B.

## 3.4 PROTEIN POCKET PREDICTION

Predicting protein binding pockets is crucial for drug design and molecular docking. We evaluate Uni-3DAR's token-level understanding on this task, formulating it as a classical atom-level classification problem where each atom is labeled as part of a pocket or not. Following previous work (Zhao et al., 2024), we train and evaluate on a composite dataset built from the CASF-2016 core set (Su et al., 2018), the PDBBind v2020 refined set (pdb, 2025), and MOAD (Hu et al., 2005). Performance is measured using the Intersection-over-Union (IoU) metric. As shown in Table 6, Uni-3DAR achieves state-of-the-art performance. Notably, it matches or exceeds specialized methods like Vabs-Net, which relies on additional features such as ESM embeddings and Solvent Accessible Surface Area, whereas Uni-3DAR uses only 3D structural information. These results highlight Uni-3DAR's strong capacity to interpret protein structures for fine-grained, atom-level prediction tasks.

## 3.5 Molecular Docking

Molecular docking, which predicts the binding pose of a ligand to a protein, is a cornerstone of drug discovery. Uni-3DAR frames this as a three-frame generation task: the first two frames are the protein and the initial ligand conformation, and the third is the predicted docked pose. We evaluate this approach on the PDBbind2020 dataset (pdb, 2025), benchmarking against 13 classical and deep learning methods using standard RMSD-based metrics (Top-1/5 success rates for RMSD $< 1$Å and $< 2$Å, and median RMSD), following the protocol of Cao et al. (2024). Uni-3DAR operates solely on atom types and coordinates, forgoing complex feature engineering and a separate scoring model; poses are ranked using the cumulative probability from the autoregressive generation. The results in Table 7 demonstrate that Uni-3DAR achieves state-of-the-art performance. It surpasses the previous best, SurfDock, on Top-1 metrics, with higher success rates for poses with RMSD $< 1$Å (44.75% vs. 40.96%) and $< 2$Å (69.06% vs. 68.41%), and a lower median RMSD (1.08Å vs. 1.18Å). While its Top-5 performance is slightly lower, likely due to its implicit scoring mechanism, these results underscore the strong potential of our unified, feature-light approach for molecular docking.

## 3.6 Molecular and Polymer Property Prediction via Pretraining

To evaluate its structure-level understanding, we assess Uni-3DAR on property prediction for small molecules and homopolymers after pretraining. For small molecules, we adopt the pretraining data, downstream tasks, and evaluation settings from state-of-the-art models Uni-Mol (Zhou et al., 2023b) and SpaceFormer (Lu et al., 2025), using Mean Absolute Error (MAE) as the metric. For homopolymers, we follow Wang et al. (2024) and use eight DFT-calculated property datasets, reporting the Root Mean Squared Error (RMSE) from a 5-fold cross-validation averaged over three seeds.

As summarized in Tables 8 and 9, Uni-3DAR demonstrates strong and versatile performance. On small molecule tasks (Table 8), it ranks first in 4 of 10 tasks and in the top two for 8 of 10, performing comparably to the specialized SpaceFormer model. On homopolymer tasks (Table 9), it ranks first in 4 of 8 tasks and in the top two for 7 of 8. These results affirm that Uni-3DAR develops robust and competitive representations for predicting properties across diverse chemical systems.

## 3.7 Additional Experiments

Due to space limitations, we present further experimental results in Appendix D. These include (1) an analysis of the benefits of unifying understanding and generation, (2) comprehensive ablation studies evaluating our proposed tokenization and MNTP, and (3) a comparison of inference speeds.

## 4 Related Work

**Octree and Hierarchical Autoregressive Models**     The coarse-to-fine hierarchical structure is widely used in 3D vision (Wang et al., 2017; Tatarchenko et al., 2017; Tang et al., 2021; Zhou et al., 2023a; Wang, 2023; Ibing et al., 2023; Zhang et al., 2024b; Ren et al., 2024). Among these works, (Ibing et al., 2023) is most similar to Uni-3DAR, as it also employs autoregressive generation using an octree. However, our method differs in three key aspects: (1) instead of relying on deep tree-level generation for fine details, we add an extra layer of fine-grained tokens to avoid excessively deep trees; (2) rather than compressing nodes via convolutional layers, we represent a compressed subtree with a single token; and (3) to handle dynamic token positions, while (Ibing et al., 2023) appends the next position to the current token, we adopt a masked next-token prediction strategy. These innovations make Uni-3DAR more efficient and effective than (Ibing et al., 2023). Recently, some image generative models have adopted a coarse-to-fine, level-by-level generation approach, such as VAR (Tian et al., 2024). Although the high-level idea appears similar, our motivation is distinct: Uni-3DAR is designed to avoid the inefficiencies of a full-size cubic grid, whereas VAR uses more tokens to boost performance. Moreover, Uni-3DAR remains within the next-token prediction framework, while VAR employs next-scale prediction.

**Microscopic 3D Structure Modeling**     Most previous generative models for microscopic 3D structures employ diffusion-based approaches (Wu et al., 2022; Anand and Achim, 2022; Hoogeboom et al., 2022; Xu et al., 2023b; Jiao et al., 2023) to generate atomic positions from noise. However, diffusion models have two major limitations. First, they require the number of atoms to be predetermined. Second, atom types are sampled from a categorical distribution, for which a proper score function is not well defined. Some studies have explored grid-based generation (O Pinheiro et al., 2024), but using a full-size 3D grid is computationally prohibitive. Other works have

investigated autoregressive models for 3D molecules (Luo and Ji, 2022; Luo et al., 2021; Zhang et al., 2025), but these models generate molecules atom by atom, requiring a predefined sequence order. For microscopic 3D structure understanding, prior studies primarily leverage SE(3)-invariant or equivariant models (Schütt et al., 2021; Fuchs et al., 2020). Additionally, unsupervised pretraining is widely used to mitigate the scarcity of labeled data (Stärk et al., 2022a; Cui et al., 2024; Yang et al., 2024; Zaidi et al., 2022; Zhou et al., 2023b). These models typically follow a BERT-style pretraining framework (Devlin, 2018), where some atoms are masked, their 3D positions are perturbed, and the model is trained to recover the ground truth. While highly effective for understanding tasks, most of these models cannot be directly applied to generation.

Some recent efforts have attempted to unify generation and understanding for microscopic data. However, most focus solely on sequence data (e.g., 1D SMILES, nucleotide sequences, or textual descriptions) and directly apply autoregressive language models (Christofidellis et al., 2023; Zhang et al., 2024a; Nguyen et al., 2024; Xia et al., 2025). While these models are straightforward, they lack essential 3D structural information, limiting their performance and applicability. Recent studies have also explored diffusion-based approaches. For example, UniGEM (Feng et al., 2024) demonstrated that a two-phase, multi-task training strategy can improve performance for both tasks. This approach combines diffusion loss with a prediction task applied during later diffusion steps. In summary, while previous work has made progress in bridging generation and understanding, Uni-3DAR is the first autoregressive framework to unify both tasks for 3D microscopic structures.

**Macroscopic 3D Structure Modeling** Macroscopic 3D structure modeling encompasses the understanding and generation of everyday objects (Chang et al., 2015; Deitke et al., 2023), scenes (Peng et al., 2023), CAD models (Wu et al., 2021; Willis et al., 2021; Xu et al., 2024), avatars (Canfes et al., 2023), and more. Similar to microscopic 3D structures, macroscopic 3D structures lack a unified representation format. Commonly used 3D representations include voxels (Wang et al., 2018b), point clouds (Xue et al., 2023), polygon meshes (Liu et al., 2024), implicit functions (Tang et al., 2021), and 3D Gaussian Splatting (Kerbl et al., 2023). Recent methods (Zhang et al., 2023a; Zhao et al., 2023; Zhang et al., 2024b; Chen et al., 2024) based on Diffusion Transformers (Peebles and Xie, 2023) encode 3D shapes into compressed, compact latent codes, substantially improving representation efficiency. Previous literature also explored autoregressive modeling for macroscopic 3D structures. For example, Polygen (Nash et al., 2020) and MeshGPT (Siddiqui et al., 2024) generate mesh faces sequentially from lowest to highest on the vertical axis, corresponding to the point-based tokenization strategy as discussed in Sec. 1, suffering from the same challenges in dynamic token positions. Another category of 3D structure generation methods, known as optimization-based approaches (Tang et al., 2023; Lin et al., 2023a; Metzer et al., 2023; Poole et al., 2022), leverages text-to-image generative models and refines 3D representations by distilling information from 2D images (Poole et al., 2022). Unlike true 3D generation, these methods primarily perform 3D reconstruction, making them fundamentally distinct from the previously mentioned 3D generation techniques and Uni-3DAR.

## 5 CONCLUSION

In this work, we introduced Uni-3DAR, a unified autoregressive framework designed to address the long-standing fragmentation of 3D modeling. By leveraging a novel coarse-to-fine octree-based tokenizer, Uni-3DAR compresses diverse 3D structures—from molecules to macroscopic shapes—into a common 1D sequence representation. This core innovation, enhanced by 2-level subtree compression for efficiency and a masked next-token prediction strategy to handle sparse spatial data, enables a single model to seamlessly bridge the gap between generative and understanding tasks across different scales. Our extensive experiments validate this unified approach, demonstrating that Uni-3DAR achieves state-of-the-art or highly competitive performance on a wide array of benchmarks. Notably, it consistently outperforms specialized, diffusion-based models while being significantly more efficient. By proving that a simple yet powerful autoregressive paradigm can unify disparate tasks without compromising accuracy, we believe Uni-3DAR marks a pivotal step toward a general-purpose foundation model (a "GPT-2 moment") for the cross-scale 3D domain.

**Limitations** While our results demonstrate the mutual benefits of unifying generation and understanding, we have not yet trained a single, large-scale foundation model on a heterogeneous mixture of 3D data and tasks. Realizing this vision through joint pretraining is a primary goal for future work. Other critical avenues for research include extending the framework to real-world applications.

## ETHICS STATEMENT

The research presented in this paper aims to advance scientific discovery by creating a unified framework for 3D modeling. We have strived to conduct this work with the highest ethical standards.

All datasets used in our experiments—including QM9, GEOM-DRUG, Materials Project, ShapeNet, and PDBBind—are publicly available and are standard benchmarks in their respective scientific communities. We did not collect any new data, and no personally identifiable or sensitive information was used.

We acknowledge that generative models for molecular and material design could potentially be misused for creating harmful substances. However, Uni-3DAR is intended as a fundamental scientific tool to accelerate beneficial research in fields such as drug discovery and materials science. Its capabilities are grounded in the principles learned from public scientific data. We believe the potential benefits—such as the rapid design of novel therapeutics and efficient materials—significantly outweigh the risks. As with any powerful technology, we advocate for its responsible use and encourage the research community to establish clear guidelines for the ethical application of generative models in science.

Finally, we recognize the environmental impact associated with training large-scale models. Our work incorporates significant efficiency optimizations, such as octree and subtree compression, which substantially reduce the computational resources and token count required for training and inference compared to alternative approaches.

## REPRODUCIBILITY STATEMENT

To ensure the reproducibility of our results, we are committed to making our research as transparent as possible.

**Code** Upon acceptance of this paper, we will release the complete source code for Uni-3DAR, including model implementation, training scripts, and evaluation protocols, under a permissive open-source license.

**Data** All datasets used in our experiments are publicly available and have been cited appropriately. We followed standard data processing and splitting protocols as established in prior work. Detailed descriptions of data preparation for each task are provided in Appendix C.

**Hyperparameters and Architecture** The full details of our model architecture, as well as the specific hyperparameters used for every experiment (including learning rates, batch sizes, model dimensions, and training steps), are thoroughly documented in Appendix B and C. We used a consistent model configuration across most tasks to demonstrate the robustness and generality of our framework.

**Computational Environment** Our experiments were conducted using standard deep learning libraries. Specific details about the hardware (e.g., NVIDIA A100 and 4090 GPUs) and software environment are provided in the appendices to facilitate the replication of our training and inference setups. The efficiency optimizations used, such as FlashAttention and KV-caching, are also described in Appendix B.

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

# Appendix

## Table of Contents

## A    THE MERITS OF USING OCTREES FOR 3D GENERATION

Octrees offer a principled way to turn sparse 3D geometry into short, informative token sequences—exactly what an autoregressive (AR) model needs to scale across domains and resolutions. Compared with uniform voxel grids and point/atom lists, an octree (i) adapts to sparsity, (ii) preserves precise spatial locality, and (iii) supplies a natural coarse-to-fine generation order that dramatically simplifies next-token prediction.

- **Token efficiency at scale.** Let the finest grid resolution be $2^L$ per axis, and let $N$ denote the number of non-empty leaf cells at level $L - 1$. A uniform $2^L \times 2^L \times 2^L$ grid yields $O(8^L)$ tokens regardless of sparsity. In contrast, an octree emits at most $8N$ tokens per level, totaling $\leq 8NL$ across $L$ levels (see Section 2.1). With our 2-level subtree compression (2LSC), a parent and its 8 children are encoded as *one* 8-bit token, reducing the count by $\approx 8\times$ to $\leq N(L-1)$. For thin structures (e.g., molecular surfaces or macroscopic shells) where $N$ scales roughly with surface area, the token complexity approaches $O(M^2 \log M)$ rather than $O(M^3)$ for a grid of side length $M = 2^L$—a decisive advantage for large systems.

- **Coarse-to-fine inductive bias.** The octree's hierarchy (Figure 2) gives each token strong context: high-level occupancy constrains where fine detail can appear, and subsequent levels specialize only within occupied regions. This bias shrinks the search space early—occupancy first, details later—so the AR model solves a sequence of easier problems rather than one monolithic one.

- **Stable, explicit positions for AR prediction.** Point- or atom-based sequences suffer from ordering ambiguities and unknown future positions. Octree nodes, however, have deterministic positions (cell centers) and levels, which we feed as positional signals. Combined with our masked next-token prediction (MNTP; Section 2.2), the model conditions on the *correct* target position before predicting content, avoiding the instability of "predict-where-then-what" pipelines.

- **Precision where it matters.** Deepening the tree only where geometry exists allocates resolution adaptively. Our fine-grained "3D patch" tokens then capture sub-voxel attributes (e.g., atom type and in-cell coordinates for molecules, or VQ codes for macroscopic shapes), marrying lossless spatial scaffolding with rich local detail (Section 2.1).

- **Small, well-posed classification tasks.** 2LSC transforms eight binary occupancy decisions into a single 256-way classification, improving statistical efficiency and reducing sequence length. Downstream heads predict small discrete/continuous targets (e.g., token type and in-cell offsets) conditioned on strong spatial priors, which is well suited to AR transformers.

- **Unified across scales and modalities.** Because the same octree scaffolding applies to Å-scale atoms and meter-scale objects, Uni-3DAR uses one tokenizer and one AR model for generation and understanding across molecules, crystals, proteins, and macroscopic shapes (Section 2.3). This uniformity simplifies conditioning (e.g., sequences, PXRD, text) and multi-frame tasks without custom architectures.

In sum, the octree representation yields shorter sequences, clearer positional signals, and a natural generation curriculum. Together with 2LSC and MNTP, it makes AR modeling practical and accurate for cross-scale 3D generation and understanding.

## B    IMPLEMENTATION DETAILS

This section outlines the technical details of our approach, covering the tokenization schemes for different scales, the model architecture, and various optimizations.

### B.1    FINE-GRAINED ATOM TOKENIZATION FOR MICROSCOPIC STRUCTURES

For microscopic 3D structures like molecules, we employ a fine-grained tokenization strategy where each token represents a single atom. This is achieved by recursively partitioning the 3D space using an octree until the final-level 3D patches are small enough to contain at most one atom. In our experiments, we set this final cell size, $c_{L-1}$, to $0.24$Å.

Each atom is thus represented by a token $(t_i, \boldsymbol{e}_i)$, where $t_i$ is the atom type (e.g., Carbon, Oxygen) and $\boldsymbol{e}_i = (e_i^0, e_i^1, e_i^2)$ specifies the atom's coordinates within its cell (we don't model the radius of a

atom). To handle continuous positions, we discretize the coordinates with a resolution $c_r = 0.01\text{Å}$, mapping them to integers in the range $\{0, \ldots, N_p - 1\}$, where $N_p = c_{L-1}/c_r$. In contrast, tokens representing non-terminal octree cells, which do not have a specific in-cell position, are assigned a default coordinate $\boldsymbol{e}_i = (N_p/2, N_p/2, N_p/2)$. For data augmentation, we apply a random rotation to the structure before tokenization.

This octree-based approach is highly efficient. For example, when applied to the QM9 dataset (Ramakrishnan et al., 2014a) using $L = 6$ levels, a typical structure with an average of 18 atoms is converted into approximately 160 tokens. This is a dramatic reduction compared to the $(2^6)^3 = 262,144$ tokens that would be required by a uniform grid of the same resolution. For othe microscopic tasks, we keep the same size $c_{L-1} = 0.24\text{Å}$ for 3D patch, while the number of levels $L$ is set according to the data type. For example, we use $L = 10$ for large proteins.

## B.2 Vector Quantized Tokenization for Macroscopic Structures

For large, macroscopic 3D structures, we adopt a voxel-based representation and employ a Vector-Quantized Variational Autoencoder (VQ-VAE) for tokenization. This approach is analogous to methods used for 2D image tokenization, where an image is converted into a sequence of discrete tokens.

The core idea is to divide a high-resolution boolean voxel grid (e.g., $512 \times 512 \times 512$) into non-overlapping 3D patches and learn a discrete, compressed representation for each one. From the input grid resolution of $512^3$ and a target latent grid of $16^3$ tokens, each token ultimately represents a $32 \times 32 \times 32$ patch of the original structure. To maintain a unified token format $(t_i, \boldsymbol{e}_i)$ with our other representations, the discrete code index from the VQ-VAE serves as the token type $t_i$, while its in-cell coordinate $\boldsymbol{e}_i$ is set to a default value.

Our VQ-VAE tokenization pipeline involves the following steps:

1. **Lossless Voxel-to-Channel Packing:** We first perform a lossless pre-processing step to make the data more amenable to standard 3D convolutional networks. Each non-overlapping $4 \times 4 \times 4$ block of the boolean input grid, containing 64 bits of information, is bit-packed into 8 bytes. This transforms the input data from a sparse, single-channel boolean tensor of shape $1 \times 512^3$ into a dense, multi-channel tensor of shape $8 \times 128^3$, where each value is an integer in $\{0, \ldots, 255\}$. This can be expressed as a mapping: $\mathbb{B}^{1 \times 512 \times 512 \times 512} \rightarrow \mathbb{U}_8^{8 \times 128 \times 128 \times 128}$.

2. **VQ-VAE Encoding:** A 3D VQ-VAE is trained on this $8 \times 128 \times 128 \times 128$ multi-channel representation. The VQ-VAE's encoder network processes this volume using a downsampling factor of 8, mapping each $8 \times 8 \times 8$ spatial patch of the multi-channel data to a single latent vector. This results in a final latent grid of $16 \times 16 \times 16$ vectors. Each vector is then quantized by finding the nearest entry in a learned codebook.

3. **Token Representation:** The output of this process is a grid of integer indices, $Z \in \{0, \ldots, N_c\}^{16 \times 16 \times 16}$, where each index corresponds to a vector in the codebook. Based on the provided code, we use a codebook with $N_c = 512$ learnable "content" codes, where each code is a vector of dimensionality $D_c = 4$.

To efficiently handle the inherent sparsity of most macroscopic structures, we introduce a special **"blank" token**. A $32 \times 32 \times 32$ patch in the original voxel grid is considered blank if and only if all voxels within it are zero. During encoding, these blank patches are mapped to a reserved index (e.g., index 0). The remaining $N_c$ indices are used for non-empty patches. This allows subsequent generative models to ignore the blank tokens, focusing computational resources exclusively on regions containing geometry. We implement this VQ-VAE using the `vector-quantize-pytorch` library, configuring it with techniques like cosine similarity, k-means initialization, and diversity losses to ensure robust codebook utilization.

## B.3 Model Architecture

We use a standard decoder-only Transformer architecture (Vaswani, 2017), based on the GPT-2 model size. The model consists of 12 layers, an embedding dimension of 768, and 12 attention heads with a head dimension of 64, totaling approximately 90M parameters. Each layer contains a

---

**Algorithm 1** A Simple Autoregressive Head for Sequential Target Prediction

---

**Require:** Input tensor $x$, number of targets $n$, prediction heads for each target $\{pred\_heads\}$, embedding layers for each prediction $\{emb\_layers\}$

1:   $y \leftarrow x$
2:   Initialize $preds \leftarrow \{\}$
3:   **for** $i \leftarrow 1$ **to** $n$ **do**
4:     $p \leftarrow pred\_heads[i](y)$
5:     Append $p$ to $preds$
6:     $y \leftarrow y + emb\_layers[i](p)$ {Teacher-forcing during training}
7:   **end for**
8:   **return** $preds$

---

unidirectional self-attention module and a SwiGLU (Shazeer, 2020) feed-forward network. For normalization, we employ a pre-norm design (Xiong et al., 2020) with RMSNorm (Zhang and Sennrich, 2019).

### B.4   INPUT EMBEDDING AND POSITIONAL ENCODING

The input representation for the $i$-th token combines several pieces of information: its type $t_i$, in-cell coordinates $e_i$, octree level $l_i$, frame index $f_i$ (for multi-frame sequences), and its absolute 3D coordinate $c_i$. For octree and masked tokens, $c_i$ is the center of the corresponding cell. For atom tokens, we use the precise atom coordinate for $c_i$ to provide a more accurate positional signal.

These discrete attributes $(t_i, e_i, l_i, f_i)$ are converted into high-dimensional vectors via separate embedding layers, and their embeddings are summed to form the final input to the model. Notably, our method does not use any 2D graphical information, such as chemical bonds, making it broadly applicable to diverse 3D data. For encoding pairwise positional information, we apply 3D Rotary Position Embedding (RoPE-3D) (Su et al., 2024) to the absolute coordinates $c_i$.

### B.5   GENERATION HEADS

The model's generative task is to predict the content of masked tokens. For an **octree token**, only the type $t_i$ needs to be predicted (since $e_i$ is fixed), which is handled by a simple classification head. For an **atom token**, both the type $t_i$ and the in-cell coordinates $e_i$ must be predicted. After predicting $t_i$, we predict $e_i$ using one of two methods:

- **Autoregressive Prediction:** The coordinates $(e_i^0, e_i^1, e_i^2)$ are predicted sequentially, as detailed in Alg. 1.
- **Diffusion Prediction:** We adapt the token-level diffusion module from MAR (Li et al., 2024b) to generate the continuous coordinates $e_i$.

Our experiments showed that both methods yield similar performance (see Sec. D.2). We therefore use the more computationally efficient autoregressive approach as our default. During inference, we employ a sampling strategy to balance quality and diversity: we first sample from the model using a slightly elevated temperature and then select the top-$r$ candidates based on their cumulative autoregressive probabilities. This method has proven more effective than standard low-temperature sampling.

### B.6   EFFICIENCY OPTIMIZATIONS

We implement several optimizations to ensure efficient training and inference. **During training**, we use FlashAttention (Dao et al., 2022) with bfloat16 to accelerate computation and reduce memory usage. We also employ sequence packing, where tokens from multiple samples are concatenated into a single sequence. This technique eliminates the overhead of padding and is particularly effective for handling systems of varying sizes, such as proteins. **During inference**, we use a KV-cache to speed up token generation. To further improve throughput for masked prediction, we generate tokens in pairs instead of one by one. This is possible because the inputs for masked tokens are known in advance, allowing us to pack adjacent prediction steps to better utilize the GPU.

Table 5: Our experiments cover a broad spectrum of real-world tasks, each of which can be seamlessly adapted by the unified framework of Uni-3DAR.

| Section | Data Type | Single-Frame Gen. | Multi-Frame Gen. | Token Und. | Structure Und. |
|---------|-----------|-------------------|------------------|------------|----------------|
| Sec. 3.1 | Molecule | ✓ | | | |
| Sec. 3.2 | Crystal + PXRD | | ✓ | | |
| Sec. 3.3 | Macroscopic 3D Object | ✓ | | | |
| Sec. 3.4 | Protein | ✓ | | ✓ | |
| Sec. 3.5 | Protein + Molecule | | ✓ | | |
| Sec. 3.6 | Molecule / Polymer | ✓ | | | ✓ |

## C  EXPERIMENT SETTINGS

### C.1  3D SMALL MOLECULE GENERATION

Generating small organic molecules with accurate 3D conformations is a classical, benchmark-rich task in molecular modeling, yet the inherent flexibility due to rotatable bonds and diverse conformations poses significant challenges. Evaluating Uni-3DAR on this task directly tests its capability to generate realistic 3D molecular structures through a straightforward application of its single-frame generation methodology.

**Dataset and Metric**  Consistent with previous studies (Hoogeboom et al., 2022), we use the QM9 (Ramakrishnan et al., 2014b) and GEOM-DRUG (Axelrod and Gomez-Bombarelli, 2022) datasets for unconditional 3D molecular generation. QM9, a widely-used molecular machine learning benchmark, contains 130K small molecules with high-quality 3D conformations (up to 9 heavy atoms and 29 total atoms including hydrogens), split into training (100K), validation (18K), and test sets (13K). GEOM-DRUG, in contrast, features larger organic compounds containing up to 181 atoms (averaging 44.2 atoms across 5 types), covering approximately 37 million conformations for around 450K unique molecules. Following established protocols (Hoogeboom et al., 2022), we select the 30 lowest-energy conformations per molecule for training.

Model performance is evaluated based on chemical feasibility. Bond types (single, double, triple, or none) are inferred from molecular geometries using pairwise atomic distances and atom types. Metrics include Atom Stability (the fraction of atoms exhibiting correct valency), Molecule Stability (the percentage of molecules where all atoms are stable), validity (percentage of chemically valid molecules verified by RDKit), and uniqueness (percentage of unique compounds among generated molecules). Metrics are computed consistently using the evaluation code from previous studies (Hoogeboom et al., 2022).

**Baselines and Implementation**  We benchmark Uni-3DAR against established models, including G-SchNet (Gebauer et al., 2022), ENF (Garcia Satorras et al., 2021), EDM (Hoogeboom et al., 2022) and its variants GDM (Hoogeboom et al., 2022), EDM-Bridge (Wu et al., 2022), GeoLDM (Xu et al., 2023b), and UniGEM (Feng et al., 2024), which uses additional molecular properties to enhance generation performance.

Uni-3DAR employs a single-frame generation approach with a batch size of 64 for QM9 and 128 for GEOM-DRUG. The model is trained for 500K steps (approximately 320 epochs for QM9 and 12 epochs for GEOM-DRUG). We apply a peak learning rate of 3e-4, incorporating a 6% linear warmup phase followed by cosine decay. Training duration is approximately 6.9 hours on 4 NVIDIA 4090 GPUs for QM9 and around 11.7 hours on 8 NVIDIA 4090 GPUs for GEOM-DRUG.

### C.2  CRYSTAL GENERATION

**Tasks**  Unlike organic molecules, crystal structures are typically rigid with stable conformations. However, crystals introduce unique challenges due to their inherent symmetry and periodic arrangement in 3D space. A crystal is conventionally represented by its lattice (a parallelepiped unit cell) along with atomic details, including atom types and their coordinates within the lattice.

In Uni-3DAR, crystal structure generation is approached as a two-frame generative process: first generating the eight vertices defining the lattice, followed by generating the atomic configurations inside the generated lattice. Notably, unlike previous methods employing fractional coordinates, we consistently use physical coordinates to maintain uniformity across various molecular data types.

Based on this generation approach, we define and address three distinct tasks:

1. *De Novo Crystal Generation*: Learning the distribution of crystal structures from data to generate novel samples unconditionally.

2. *Crystal Structure Prediction (CSP)*: Predicting crystal structures from given chemical compositions (atom types and counts). During inference, the chemical composition is provided as condition, enabling the model to generate the corresponding crystal structure.

3. *PXRD-guided Crystal Structure Prediction*: Establishing a cross-modal mapping from powder X-ray diffraction (PXRD) signals and chemical compositions to reconstruct crystal structures that accurately match observed PXRD patterns. This task has significant practical implications, as PXRD analysis is widely used in crystal structure determination and validation of novel materials in real-world scenarios.

**Dataset and Metric**  We employ established datasets consistent with prior studies (Xie et al., 2021; Jiao et al., 2023; Miller et al., 2024) for both training and evaluation purposes. Specifically, we employ the Carbon-24 dataset (Pickard, 2020), containing 10,153 carbon-based structures with cells composed of 6 to 24 atoms. The MP-20 dataset (Jain et al., 2013), derived from the Materials Project (Jain et al., 2013), includes 45,231 stable inorganic materials representing a wide range of experimentally validated compounds, each containing up to 20 atoms per cell. Additionally, we use the more challenging MPTS-52 dataset, an extended version of MP-20, comprising 40,476 structures with up to 52 atoms per cell, organized by the earliest publication year. We follow the same data split strategy as outlined in previous work (Jiao et al., 2023).

To evaluate de novo crystal generation performance, we adopt the standard evaluation framework proposed by Xie et al. (2021), which includes three key metrics: validity, coverage, and property statistics. Validity quantifies the proportion of generated structures that satisfy established physical plausibility criteria. Coverage measures the ability of generated structures to capture the diversity present in the test set. Property statistics compare essential attributes such as density, formation energy, and elemental composition between generated and ground-truth distributions.

For assessing performance in CSP and PXRD-guided CSP tasks, we align our evaluation methodology with prior research (Miller et al., 2024). We compute the top-1 match rate alongside the corresponding average root-mean-square error (RMSE) for matched structures. We employ StructureMatcher(Ong et al., 2013), using thresholds set to stol=0.5, angle_tol=10, and ltol=0.3, consistent with the methodology of previous studies (Miller et al., 2024).

**Baseline Models and Implementation**  We benchmark Uni-3DAR against established methods, including FTCP (Ren et al., 2021), G-SchNet (Gebauer et al., 2019), P-G-SchNet (Gebauer et al., 2019), CDVAE (Xie et al., 2021), DiffCSP (Jiao et al., 2023), and FlowMM (Miller et al., 2024). Additionally, we evaluate Uni-3DAR against the recent UniGenX (Zhang et al., 2025) for the CSP task. For PXRD-guided CSP, we compare Uni-3DAR with PXRDGEN (Li et al., 2024a), a model tailored for this task.

In Uni-3DAR, we use a 12-layer model with a 768-dimensional embedding for de novo crystal generation, while a larger 24-layer model with a 1024-dimensional embedding is employed for CSP and PXRD-guided CSP tasks. All models are trained for 400k steps with a batch size of 64 and a peak learning rate of 3e-4. For chemical composition conditioning, we prepend a token derived from a multi-hot atom-type vector. PXRD data, spanning angles from $0°$ to $120°$, is converted into a 1200-dimensional vector with a $0.1°$ resolution, evenly divided into four segments, each represented by a conditional token. As a result, PXRD-guided CSP utilizes a total of five conditional tokens (one for composition and four for PXRD signals). The autoregressive nature of Uni-3DAR enables seamless integration of these conditional tokens, eliminating the need for additional encoders required by previous methods (Li et al., 2024a; Lai et al., 2025).

**Results of De Novo Crystal Generation**   The performance of Uni-3DAR on the Carbon-24 and MP-20 datasets is presented in Table 2. On Carbon-24, Uni-3DAR outperforms existing models, particularly excelling in coverage, demonstrating its ability to generate diverse and realistic structures. On MP-20, Uni-3DAR significantly enhances component validity compared to previous approaches while maintaining competitive performance on other metrics. These results underscore Uni-3DAR's strength in producing chemically valid crystal structures that closely align with key physical and chemical properties.

**Results of Crystal Structure Prediction (CSP)**   We evaluate Uni-3DAR's performance on CSP across all datasets, as summarized in Table 3. Uni-3DAR consistently outperforms baseline methods by significant margins. Specifically, on Carbon-24, it improves the match rate by 4.14% over the previous best method, demonstrating superior accuracy in reconstructing crystal structures. On MP-20, Uni-3DAR achieves a substantial improvement in RMSE, reducing it from 0.0566 to 0.0317, a relative improvement of 178% over the second-best model. Furthermore, on MPTS-52, Uni-3DAR achieves an impressively low RMSE of 0.0684, representing a 184% relative improvement, despite the increased structural complexity. This result highlights its exceptional precision in atomic placement. Overall, these findings demonstrate Uni-3DAR's strong generalization capability across datasets of varying difficulty levels.

**Results of PXRD-Guided CSP**   Table 3 demonstrates Uni-3DAR's performance in PXRD-guided CSP on the MP-20 dataset, benchmarked against PXRDGEN (Li et al., 2024a). Uni-3DAR substantially outperforms PXRDGEN, elevating the match rate from 68.68% to 75.08% while drastically reducing the RMSE from 0.0707 to 0.0276—a 256% relative improvement. This significant RMSE reduction underscores Uni-3DAR's exceptional ability to generate crystal structures that precisely correspond to experimental PXRD patterns. Collectively, these results underscore the superior capability of Uni-3DAR in harnessing diffraction constraints to reliably predict crystal structures.

## C.3 MACROSCOPIC 3D OBJECT GENERATION

To demonstrate versatility beyond the microscopic realm of molecules and crystals, we further evaluate Uni-3DAR on unconditional macroscopic 3D object generation, a core task in 3D computer vision (3DCV). The goal is to synthesize realistic and diverse 3D shapes of everyday objects directly from the learned data distribution.

**Dataset and Evaluation Protocol**   Following common practice, we adopt three categories from ShapeNet (Chang et al., 2015)—*airplane*, *chair*, and *car*. Each object is represented as a point cloud with 2,048 points uniformly sampled from the surface. In line with recent recommendations, we evaluate using **1-nearest-neighbor accuracy (1-NNA; lower is better)** computed with both Chamfer Distance (CD) and Earth Mover's Distance (EMD) (Yang et al., 2019; Vahdat et al., 2022). Concretely, given a generated set $S_g$ and a reference set $S_r$, 1-NNA is the leave-one-out accuracy of a 1-NN classifier on $S_g \cup S_r$; if $S_g$ matches $S_r$ well, the classification accuracy approaches 50%. Compared with legacy metrics such as coverage (COV) and minimum matching distance (MMD), 1-NNA more directly captures distributional similarity while jointly reflecting both quality and diversity, and avoids several known failure modes of COV/MMD. We therefore report 1-NNA (with CD/EMD) as our primary metric throughout this section and in the main paper.

**Baselines and Implementation Details**   We benchmark Uni-3DAR against established point-cloud generative models, including r-GAN and l-GAN (Achlioptas et al., 2018), PointFlow (Yang et al., 2019), SoftFlow (Kim et al., 2020), SetVAE (Kim et al., 2021), DPF-Net (Klokov et al., 2020), diffusion-based methods DPM (Luo and Hu, 2021) and PVD (Zhou et al., 2021), and the recent LION (Vahdat et al., 2022). To ensure clear and reproducible comparisons, we follow the PointFlow data protocol and training/test splits for the three categories.

For Uni-3DAR, the input 3D object is voxelized at $512 \times 512 \times 512$ resolution. We define fine-grained structural tokens as non-overlapping $16 \times 16 \times 16$ voxel patches and quantize each patch with a VQVAE codebook. (In our main text we summarize this as "each patch is quantized using VQVAE"; here we provide the fuller setup for completeness.) Unless otherwise specified, generation uses our single-frame sampling procedure, analogous to the molecular setting. For each ShapeNet category, the VQVAE is trained for 200 epochs; Uni-3DAR is then trained for 10,000 steps with

Table 6: Results for atom-level binding site prediction measured by IoU (%). Baseline results are taken from Zhao et al. (2024). For a fair comparison with other methods, we report Vabs-Net's result using only $\alpha$-carbon atoms.

| Method | pretrained | B277↑ | DT198↑ | ASTEX85↑ | CHEN251↑ | COACH420↑ |
|---|---|---|---|---|---|---|
| FPocket (Le Guilloux et al., 2009) | × | 31.5 | 23.2 | 34.1 | 25.4 | 30.0 |
| SiteHound (Hernandez et al., 2009) | × | 36.4 | 23.1 | 38.9 | 29.4 | 34.9 |
| MetaPocket2 (Macari et al., 2019) | × | 37.3 | 25.8 | 37.5 | 32.8 | 37.7 |
| DeepSite (Jiménez et al., 2017) | × | 34.0 | 29.1 | 37.4 | 27.4 | 33.9 |
| P2Rank (Krivák and Hoksza, 2018) | × | 49.8 | 38.6 | 47.4 | **56.5** | 45.3 |
| ESM2_150M (Lin et al., 2023b) | √ | 19.6 | 16.6 | 20.5 | 18.9 | 22.0 |
| GearNet (Zhang et al., 2022b) | √ | 39.9 | 35.8 | 41.0 | 36.4 | 41.3 |
| Siamdiff (Zhang et al., 2023d) | √ | 37.7 | 31.0 | 40.7 | 35.3 | 40.3 |
| Vabs-Net (Zhao et al., 2024) | √ | - | - | - | - | **56.3** |
| Uni-3DAR | √ | **53.4** | **46.7** | **51.4** | 47.9 | 56.2 |

batch size 64. On a single NVIDIA RTX 4090, training per category requires approximately 10 hours for the VQVAE and 2 hours for Uni-3DAR.

**Results**   Table 4 (main paper) summarizes unconditional generation under the 1-NNA protocol. Uni-3DAR achieves the lowest (best) 1-NNA in all six category–metric pairs (Airplane/Chair/Car × CD/EMD), outperforming strong diffusion and flow-based baselines. In particular, Uni-3DAR consistently improves over LION—the strongest baseline in our comparison—by small but systematic margins: *Airplane* (CD: 67.35 vs. 67.41; EMD: 61.09 vs. 61.23), *Chair* (CD: 53.11 vs. 53.70; EMD: 50.98 vs. 52.34), and *Car* (CD: 53.35 vs. 53.41; EMD: 50.89 vs. 51.14). Taken together, these results indicate that Uni-3DAR produces point-cloud distributions that are both high-quality and diverse, closely matching the real data according to a metric expressly designed to assess distributional similarity.

## C.4   PROTEIN POCKET PREDICTION

Proteins are a crucial class of biological structures, and accurate prediction of binding pockets is essential for de novo drug design and applications such as molecular docking. Traditionally, pocket prediction is formulated as an atom-level or residue-level classification task. Each atom or residue is assigned a binary label indicating whether it belongs to a binding pocket. We adopt this classical formulation to evaluate Uni-3DAR's token-level understanding capabilities.

**Dataset and Metric**   We follow previous studies (Zhao et al., 2024) and employ a binding site dataset constructed from the CASF-2016 core set (Su et al., 2018), PDBBind v2020 refined set (pdb, 2025), and MOAD (Hu et al., 2005). The dataset consists of 23k training samples, 5k validation samples, and five test sets of roughly 1k samples each. Model performance is assessed using the Intersection-over-Union (IoU) metric, consistent with previous evaluations (Zhao et al., 2024).

**Baselines and Implementation**   We benchmark Uni-3DAR against established methods. Our comparisons include non-pretrained approaches (e.g., FPocket (Le Guilloux et al., 2009), SiteHound (Hernandez et al., 2009), etc.) and pretrained models (e.g., ESM2_150M (Lin et al., 2023b), GearNet (Zhang et al., 2022b), Siamdiff (Zhang et al., 2023d), and Vabs-Net (Zhao et al., 2024)). In line with prior works (Zhao et al., 2024), we pretrain Uni-3DAR on approximately 1.3 million protein structures before fine-tuning it on the binding site dataset. Unlike Vabs-Net, which employs full-atom representations, our experiments are restricted to $\alpha$-carbon atoms to facilitate direct comparisons.

Pretraining is conducted using a single-frame generation approach for 300k steps with a batch size of 64. We use a peak learning rate of 3e-4 with a 10% linear warmup followed by cosine decay, which requires approximately 19 hours on 16 NVIDIA A100 GPUs. Fine-tuning adopts an atom-level classification strategy, conducted for 100 epochs with a batch size of 32, a peak learning rate of 1e-4, requiring roughly 7 hours on 8 NVIDIA A100 GPUs.

Table 7: Comparison of docking performance on the Top1- and Top5-RMSD metrics. The first group of five baselines comprises classical docking software, while the second group of eight baselines consists of deep learning–based methods. The results are reproduced directly from Cao et al. (2024). The best outcomes are shown in **bold**, and the second-best are underlined.

| | Top1-RMSD | | | Top5-RMSD | | |
|---|---|---|---|---|---|---|
| | %<1Å ↑ | %<2Å ↑ | Med(Å) ↓ | %<1Å ↑ | %<2Å ↑ | Med(Å) ↓ |
| Uni-Dock (Yu et al., 2022) | 32.51±0.39 | 50.69±0.59 | 1.89±0.04 | 47.11±0.22 | 67.03±0.94 | 1.10±0.02 |
| Glide SP (Friesner et al., 2004) | 17.36±0.00 | 44.63±0.00 | 2.27±0.00 | 31.13±0.00 | 60.06±0.00 | 1.54±0.00 |
| GNINA (Ragoza et al., 2017) | 21.12±0.26 | 43.62±1.06 | 2.45±0.07 | 28.47±0.57 | 58.13±0.81 | 1.65±0.02 |
| SMINA (Koes et al., 2013) | 18.73±0.00 | 31.68±0.00 | 3.99±0.00 | 28.47±0.56 | 48.48±0.00 | 2.07±0.00 |
| Vina (Eberhardt et al., 2021) | 18.32±0.02 | 36.64±0.05 | 3.42±0.01 | 24.79±0.00 | 50.96±0.00 | 1.87±0.01 |
| EquiBind (Stärk et al., 2022b) | / | 5.5±1.2 | 6.2±0.3 | / | / | / |
| TANKBind (Lu et al., 2022) | 2.66±0.26 | 18.18±0.60 | 4.2±0.05 | 4.13±0.0 | 20.39±0.45 | 3.5±0.04 |
| E3Bind (Zhang et al., 2022a) | / | 25.6 | 7.2 | / | / | / |
| KarmaDock (Zhang et al., 2023c) | / | 56.2 | / | / | / | / |
| DiffDock(Pocket) (Corso et al.) | / | 51.8 | 2.0 | / | 60.7 | 1.9 |
| DiffDock (Corso et al.) | 15.15 | 36.09 | 3.35 | 21.76 | 43.52 | 2.46 |
| DiffDock-L (Corso et al., 2024) | 19.07±0.57 | 40.74±1.25 | 2.88±0.18 | 21.95±0.39 | 48.15±0.91 | 2.05±0.04 |
| SurfDock (Cao et al., 2024) | 40.96±0.34 | 68.41±0.26 | 1.18±0.00 | 54.18±0.13 | **75.11**±0.13 | 0.94±0.00 |
| Uni-3DAR | **44.75**±2.63 | **69.06**±0.75 | **1.08**±0.04 | **56.35**±1.99 | 72.38±0.73 | **0.76**±0.02 |

## C.5 MOLECULAR DOCKING

Molecular docking predicts how a ligand binds to a target protein, playing a crucial role in drug discovery. In Uni-3DAR, this process is structured as a three-frame generation task. The first two frames represent the protein and the initial ligand, both provided as inputs during inference, while the third frame corresponds to the predicted docked conformation of the ligand.

**Dataset and Metric**    Following Cao et al. (2024), we train and evaluate docking methods on the PDBbind2020 dataset. The training and validation set consists of 17,000 complexes from 2018 or earlier, while the test set includes 363 structures from 2019, ensuring no ligand overlap with the training data. Given a protein-binding pocket and a randomly generated ligand conformation from RDKit, the goal is to generate a user-specified number of poses (set to 40, as in Cao et al. (2024)). Docking methods typically incorporate a confidence scoring mechanism to rank these poses. Performance is assessed using the percentage of predictions with RMSD < 1Å and RMSD < 2Å, as well as the median RMSD for the top-ranked pose and the best pose among the top five ranked poses.

**Baselines and Implementation**    We evaluate Uni-3DAR against 13 baselines, including five classical docking software tools and eight deep learning–based methods. Most existing deep learning approaches rely on complex featurizations, such as using protein language model embeddings (e.g., from ESM2 (Lin et al., 2022)). To simplify and unify the molecular tasks, we omit these complicated features in Uni-3DAR and instead use only atom types and coordinates. We also adopt a full-atom representation of the protein pocket to enhance expressive power. We frame docking as an autoregressive generation task by embedding both the pocket and the RDKit conformation as two frames, concatenating them into a single input sequence, and training the model to generate the docked molecule conformation as a new frame sequence. For further simplicity, we do not impose constraints such as matching the number and types of atoms in the output frame to those of the input molecule. Also, we do not train a separate scoring model for pose ranking. Instead, we use the cumulative probability derived from autoregressive generation to score each generated pose. We train Uni-3DAR for 300k steps (approximately 300 epochs) with a batch size of 16. The learning rate schedule follows the same configuration as the experiments detailed in section 3.1. The training is completed in approximately one day on 4 NVIDIA A100 GPUs.

**Results**    Experimental results are summarized in Table 7. Uni-3DAR outperforms to the state-of-the-art method, SurfDock, demonstrating similar percentages of poses with RMSD below 1Å and 2Å. Notably, Uni-3DAR excels in generating higher-quality poses, reflected by its lower median

Table 8: Results on molecular property prediction performance. The best results are highlighted in **bold**, and the second-best results are underlined. Baseline results are taken from Lu et al. (2025).

| Model | HOMO↓ (Hartree) | LUMO↓ (Hartree) | GAP↓ (Hartree) | E1-CC2↓ (eV) | E2-CC2↓ (eV) | f1-CC2↓ | f2-CC2↓ | Dipmom↓ (Debye) | aIP↓ (eV) | D3_disp↓ _corr (eV) |
|---|---|---|---|---|---|---|---|---|---|---|
| GROVER (Rong et al., 2020) | 0.0075 ± 2.0e-4 | 0.0086 ± 8.0e-4 | 0.0109 ± 1.4e-3 | 0.0101 ± 9.7e-4 | 0.0129 ± 4.6e-4 | 0.0219 ± 3.5e-4 | 0.0401 ± 1.2e-3 | 0.0752 ± 1.1e-3 | 0.1467 ± 1.5e-2 | 0.2516 ± 5.3e-2 |
| GEM (Fang et al., 2022) | 0.0068 ± 7.0e-5 | 0.0080 ± 2.0e-5 | 0.0107 ± 1.9e-5 | 0.0090 ± 1.3e-4 | 0.0102 ± 2.3e-4 | 0.0170 ± 4.3e-4 | 0.0352 ± 5.4e-4 | 0.0289 ± 1.2e-3 | 0.0207 ± 2.6e-4 | 0.0077 ± 6.6e-4 |
| 3D Infomax (Stärk et al., 2022a) | 0.0065 ± 1.0e-5 | 0.0070 ± 1.0e-4 | 0.0095 ± 1.0e-4 | 0.0089 ± 2.0e-4 | 0.0091 ± 3.0e-4 | 0.0172 ± 4.0e-4 | 0.0364 ± 9.0e-4 | 0.0291 ± 1.7e-3 | 0.0526 ± 1.4e-4 | 0.2285 ± 7.5e-3 |
| Uni-Mol (Zhou et al., 2023b) | 0.0052 ± 2.0e-5 | 0.0060 ± 6.0e-5 | 0.0081 ± 4.0e-5 | 0.0067 ± 4.0e-5 | 0.0080 ± 4.0e-5 | 0.0143 ± 2.0e-4 | 0.0309 ± 9.4e-4 | 0.0106 ± 3.1e-4 | 0.0095 ± 6.4e-4 | **0.0047** ± 5.6e-4 |
| Mol-AE (Yang et al., 2024) | 0.0050 ± 8.0e-5 | 0.0057 ± 4.7e-4 | 0.0080 ± 8.0e-5 | 0.0070 ± 6.0e-5 | 0.0080 ± 4.0e-5 | 0.0140 ± 4.0e-5 | 0.0307 ± 1.3e-3 | 0.0113 ± 4.7e-4 | 0.0103 ± 1.3e-4 | 0.0077 ± 1.3e-3 |
| SpaceFormer (Lu et al., 2025) | **0.0042** ± 1.0e-5 | **0.0040** ± 2.0e-5 | **0.0064** ± 1.2e-4 | 0.0058 ± 8.0e-5 | 0.0074 ± 8.4e-5 | 0.0142 ± 3.7e-4 | 0.0294 ± 7.1e-4 | **0.0083** ± 5.0e-4 | **0.0090** ± 5.9e-4 | 0.0053 ± 1.2e-3 |
| Uni-3DAR | 0.0048 ± 2.1e-5 | 0.0044 ± 3.2e-5 | 0.0065 ± 8.8e-5 | **0.0056** ± 2.2e-5 | **0.0067** ± 2.0e-5 | **0.0134** ± 7.0e-5 | **0.0286** ± 1.6e-4 | 0.0114 ± 6.9e-4 | 0.0127 ± 1.1e-4 | 0.0052 ± 3.2e-4 |

RMSD values. However, Uni-3DAR exhibits slightly inferior performance in selecting Top-5 poses for challenging cases, as evidenced by a lower percentage of poses with RMSD below 2Å in the Top5-RMSD evaluation (72.38% vs. 75.11% for SurfDock). This gap may arise because the scoring module in Uni-3DAR has not been explicitly trained and is only exposed to ground-truth conformations during the training phase. Addressing this limitation by training a dedicated scoring module could potentially enhance its selection performance. Moreover, since Uni-3DAR avoids complex feature engineering, its docking accuracy might further benefit from multitask learning strategies, emphasizing the promise of a unified foundational model for molecular applications.

## C.6 MOLECULAR PROPERTY PREDICTION VIA PRETRAINING

Molecular property prediction through pretraining has emerged as an effective strategy to address data scarcity challenges in areas like drug discovery and material design. As a classical task with established benchmarks, molecular property prediction directly assesses a model's capacity to comprehend 3D molecular structures. Applying Uni-3DAR's structure-level understanding framework is thus straightforward.

**Dataset and Metric**   We utilize the same pretraining dataset as employed by Uni-Mol (Zhou et al., 2023b) and SpaceFormer (Lu et al., 2025), comprising approximately 19 million molecules. For downstream evaluations, we follow the datasets and evaluation settings used by the state-of-the-art SpaceFormer (Lu et al., 2025). These include a 20K dataset predicting electronic properties (HOMO, LUMO, GAP), a 21K dataset targeting energy properties (E1-CC2, E2-CC2, f1-CC2, f2-CC2), and an 8K dataset predicting mechanical and electronic properties (Dipmom, aIP, and D3 Dispersion Corrections). Data splits align exactly with SpaceFormer's methodology (Lu et al., 2025). Performance across all tasks is measured using the Mean Absolute Error (MAE) metric.

**Baselines and Implementation**   Our baselines encompass several prominent models, including Uni-Mol (Zhou et al., 2023b), Mol-AE (Yang et al., 2024), 3D Infomax (Stärk et al., 2022a), GROVER (Rong et al., 2020), GEM (Fang et al., 2022), and the most recent state-of-the-art method, SpaceFormer (Lu et al., 2025). For pretraining, we use the proposed masked next-token prediction as pretraining task, training the model for 500k steps with a batch size of 128. The peak learning rate is set to 3e-4, incorporating a 10% linear warmup followed by cosine decay, requiring approximately 11.5 hours on 8 NVIDIA 4090 GPUs.

During fine-tuning, we adopt a structure-level understanding strategy, supplemented by a masked next-token prediction auxiliary generative loss. Training is conducted over a maximum of 200 epochs. We systematically explore hyperparameter combinations, considering two batch sizes (32, 64) and two learning rates (5e-4, 1e-4), resulting in four distinct setups. For each hyperparameter configuration, models are trained three times using different random seeds, and we report the mean performance along with standard deviation. The best-performing model based on validation loss is selected for evaluation.

Table 9: Polymer properties prediction performance. The best results are highlighted in **bold**, and the second-best results are underlined.

| Model | Egc ↓ (eV) | Egb ↓ (eV) | Eea ↓ (eV) | Ei ↓ (eV) | Xc ↓ % | Eps ↓ 1 | Nc ↓ 1 | Eat ↓ eV/atom |
|---|---|---|---|---|---|---|---|---|
| ChemBERTa (Chithrananda et al., 2020) | 0.539 ± 0.049 | 0.664 ± 0.079 | 0.350 ± 0.036 | 0.485 ± 0.086 | 18.711 ± 1.396 | 0.603 ± 0.083 | 0.140 ± 0.010 | 0.219 ± 0.056 |
| Uni-Mol (Zhou et al., 2023b) | 0.489 ± 0.028 | 0.531 ± 0.055 | 0.332 ± 0.027 | 0.407 ± 0.080 | 17.414 ± 1.581 | 0.536 ± 0.053 | 0.095 ± 0.013 | 0.084 ± 0.034 |
| SML (Zhang et al., 2023b) | 0.489 ± 0.056 | 0.547 ± 0.110 | 0.313 ± 0.016 | 0.432 ± 0.060 | 18.981 ± 1.258 | 0.576 ± 0.020 | 0.102 ± 0.010 | 0.062 ± 0.014 |
| PLM (Zhang et al., 2023b) | 0.459 ± 0.036 | 0.528 ± 0.081 | 0.322 ± 0.037 | 0.444 ± 0.062 | 19.181 ± 1.308 | 0.576 ± 0.060 | 0.100 ± 0.010 | **0.050** ± 0.010 |
| polyBERT (Kuenneth and Ramprasad, 2023) | 0.553 ± 0.011 | 0.759 ± 0.042 | 0.363 ± 0.037 | 0.526 ± 0.068 | 18.437 ± 0.560 | 0.618 ± 0.049 | 0.113 ± 0.003 | 0.172 ± 0.016 |
| Transpolymer (Xu et al., 2023a) | 0.453 ± 0.007 | 0.576 ± 0.021 | 0.326 ± 0.040 | 0.397 ± 0.061 | 17.740 ± 0.732 | 0.547 ± 0.051 | 0.096 ± 0.016 | 0.147 ± 0.093 |
| MMPolymer (Wang et al., 2024) | 0.431 ± 0.017 | 0.503 ± 0.038 | **0.286** ± 0.029 | **0.390** ± 0.057 | **16.814** ± 0.867 | 0.511 ± 0.035 | **0.087** ± 0.010 | 0.061 ± 0.016 |
| Uni-3DAR | **0.426** ± 0.022 | **0.498** ± 0.048 | 0.291 ± 0.022 | 0.396 ± 0.072 | 17.16 ± 1.498 | **0.487** ± 0.034 | **0.087** ± 0.011 | 0.066 ± 0.031 |

## C.7 POLYMER PROPERTY PREDICTION VIA PRETRAINING

Polymers, synthesized through various polymerization methods such as addition, ring-opening, and condensation, consist of repeating monomer units. These materials play essential roles across multiple fields, including materials science, drug design, and bioinformatics, necessitating accurate property prediction methods. Here, we demonstrate Uni-3DAR's structure-level understanding capability by focusing on homopolymer property prediction.

**Dataset and Metric** Following prior research (Zhang et al., 2023b; Wang et al., 2024), we use eight publicly available polymer property datasets (Egc, Egb, Eea, Ei, Xc, EPS, Nc, and Eat), obtained via density functional theory (DFT) calculations. Given that all tasks involve structure-level regression, we employ a robust evaluation strategy using 5-fold cross-validation with random splits, consistent with previous work (Wang et al., 2024). Results are reported as the root mean squared error (RMSE), averaged across three different random seeds.

**Baselines and Implementation** Baseline methods include ChemBERTa (Chithrananda et al., 2020), Uni-Mol (Zhou et al., 2023b), SML (Zhang et al., 2023b), PML (Zhang et al., 2023b), poly-BERT (Kuenneth and Ramprasad, 2023), Transpolymer (Xu et al., 2023a), and MMPolymer (Wang et al., 2024). For pretraining, we represent homopolymers as specialized molecular structures using the star substitution strategy proposed in (Wang et al., 2024). The model is pretrained using our masked next-token prediction strategy for 1 million steps with a batch size of 128. All other experimental details follow the settings previously described in the molecular property prediction experiments.

During fine-tuning, we adopt structure-level understanding strategy with masked next-token prediction auxiliary generative loss. Training is capped at 200 epochs. We thoroughly investigate various hyperparameter combinations by using three different batch sizes (32, 64, 128) and three learning rates (5e-4, 1e-4, 3e-4), creating nine unique configurations. Each configuration is tested by training models three times with different random seeds. We follow the same 5-fold split index align with (Wang et al., 2024), by averaging the best validation metrics in each fold. Subsequently, we present the mean performance along with the standard deviation across three seeds.

## D MORE EXPERIMENTS

### D.1 MUTUAL BENEFITS OF GENERATION AND UNDERSTANDING TASKS

In the previous experiments, we applied Uni-3DAR independently to each task to ensure fair comparisons with established approaches, rather than employing joint training across multiple tasks and diverse data sources. Although earlier results already demonstrate Uni-3DAR's effectiveness, the

Table 10: In the molecular pretrained representation task, incorporating a generation loss during downstream fine-tuning improves performance.

|  | HOMO ↓ (Hartree) | LUMO ↓ (Hartree) | E1-CC2 ↓ (eV) | E2-CC2 ↓ (eV) |
|---|---|---|---|---|
| Uni-3DAR w/o Gen. loss | 0.0052 | 0.0049 | 0.0063 | 0.0077 |
| Uni-3DAR | **0.0048** | **0.0044** | **0.0056** | **0.0067** |

Table 11: In the QM9 unconditional generation task, incorporating a structure-level understanding task further enhances the quality of the generated samples.

|  | QM9 | | | |
|---|---|---|---|---|
|  | Atom Sta(%)↑ | Mol Sta(%)↑ | Valid(%)↑ | V × U(%)↑ |
| Uni-3DAR | 99.4 | 93.7 | 98.0 | **94.0** |
| Uni-3DAR w/ Structure Und. loss | **99.6** | **95.8** | **98.5** | 93.1 |

advantages of joint training, particularly combining generation and understanding tasks, remain less explored. Due to resource limitations, comprehensive large-scale joint training was not feasible in this paper. Nonetheless, this subsection presents two additional experiments that clearly illustrate how generation and understanding tasks can mutually reinforce each other, highlighting the potential for enhanced performance through joint training in Uni-3DAR.

The first experiment leverages the pretrained molecular representation described in Sec. 3.6. Typically, during downstream fine-tuning, we include an auxiliary generative loss by predicting ground-truth atom types and positions with the proposed masked next-token prediction. To investigate the contribution of this auxiliary generation task, we performed an ablation experiment by removing the generation loss during fine-tuning (results shown in Table 10). The results indicate a notable performance drop without the generation loss, clearly demonstrating that generative training significantly strengthens structure-level understanding.

The second experiment builds upon the unconditional 3D molecule generation task using the QM9 dataset described in Sec. 3.1. Previously, to align with prior studies, we used only 3D molecular structure data. Here, we additionally incorporate a structure-level understanding task by predicting the molecular property $U$ (internal energy at 298.15 K), with results shown in Table 11. Models trained with this auxiliary structure-level understanding task consistently outperform those without, especially in metrics such as molecular stability and validity. This demonstrates that structure-level understanding significantly enhances generative performance.

In summary, these experiments robustly illustrate that generation and understanding tasks positively reinforce one another. The findings underscore that integrating diverse datasets and joint task training can establish a more powerful and effective foundation model for 3D structural modeling.

## D.2   ABLATION STUDY

We conducted comprehensive ablation experiments on QM9 generation task (Sec. 3.1) to evaluate the contributions of key components in Uni-3DAR. The experimental results, summarized in Table 12, lead to the following insights:

1. **Masked Next-Token Prediction significantly enhances generation performance.** In experiment No.2, we followed previous work (Ibing et al., 2023) that merely appends the position of the next token to the current token, without using our proposed masked next-token prediction. Comparing experiments No.1 and No.2 clearly demonstrates that our proposed masked next-token prediction substantially outperforms this baseline approach.

2. **2-Level Subtree Compression boosts efficiency without compromising performance.** Experiment No.3 evaluates performance without 2-level subtree compression. Comparing No.1 (with compression) and No.3 (without compression), we observe that using subtree compression reduces token count by approximately 6x, leading to significantly faster training with comparable results. Interestingly, experiment No.4 (No.2 without subtree compression) outperforms No.2. This indicates that while subtree compression alone may slightly impact performance negatively

Table 12: Ablation Studies for Uni-3DAR. MNTP (Masked Next-Token Prediction) boosts performance, while 2LSC (2-Level Subtree Compression) enhances efficiency. Uni-3DAR integrates both techniques to balance effectiveness and efficiency. Token-level diffusion loss (diff. loss) performs comparably to our proposed simple autoregressive head. Training cost is measured using 4 NVIDIA 4090 GPUs.

| No. | Settings | QM9 | | | | # AVG. | Training |
|-----|----------|-----------|-----------|----------|-------------------|--------|----------|
|     |          | Atom Sta(%)↑ | Mol Sta(%)↑ | Valid(%)↑ | V × U(%)↑ | Tokens | Cost ↓ |
| 1 | Uni-3DAR | 99.4 | 93.7 | 98.0 | 94.0 | 160 | 6.9h |
| 2 | 1 w/o MNTP | 98.7 | 88.2 | 97.0 | 91.5 | 80 | 6h |
| 3 | 1 w/o 2LSC | 99.4 | 94.4 | 98.2 | 92.1 | 1060 | 20h |
| 4 | 2 w/o 2LSC | 99.3 | 94.2 | 97.7 | 92.7 | 530 | 11h |
| 5 | 2 w/o octree | 87.7 | 25.3 | 72.1 | 65.7 | 18 | 3h |
| 6 | 1 w/ diff. loss | 99.4 | 93.6 | 98.2 | 94.0 | 160 | 7.8h |
| 7 | 5 w/ diff. loss | 88.3 | 35.4 | 67.3 | 46.5 | 18 | 5.6h |

(No.2 vs. No.4), when combined with masked next-token prediction (No.1 vs. No.3), it achieves comparable performance efficiently.

3. **Coarse-to-fine octree tokens provide essential spatial information.** In experiment No.5, we removed octree tokens, significantly degrading model performance. Without coarse-to-fine tokenization, the model degrades to atom-based autoregressive prediction of both atom types and positions, a much more challenging task. Our coarse-to-fine octree tokenization method effectively provides positional priors from preceding levels, substantially enhancing performance. This clearly validates the importance of the coarse-to-fine tokenization strategy for 3D structural generation.

4. **Token-level diffusion loss yields comparable performance to the autoregressive head but with lower efficiency.** Our default generation head uses a simple autoregressive head (refer to Alg. 1) to sequentially predict atom types and in-cell positions. We examined whether employing a more powerful head, such as the token-level diffusion loss from MAR (Li et al., 2024b), could further enhance performance. Experiment No.6, utilizing the diffusion head, achieved similar results but required more computational time. Therefore, we opt for the simpler, more efficient autoregressive head by default.

5. **Combining atom-based autoregressive and diffusion losses without spatial tokenization is insufficient.** Recent works have explored improving atom-based autoregressive generation through token-level diffusion losses (Zhang et al., 2025). We tested this approach by adding a token-level diffusion loss to experiment No.5, resulting in experiment No.7. Although No.7 performed slightly better than No.5, it remained significantly inferior to the proposed Uni-3DAR. This underscores that comprehensive spatial information, as provided by our tokenization strategy, is crucial, mere integration of diffusion-based methods into atom-based autoregressive model, without spatial tokenization, cannot achieve substantial performance improvements.

D.3 INFERENCE SPEED

We benchmarked Uni-3DAR against the diffusion-based generative model GeoLDM (Xu et al., 2023b) on QM9 generation task (Sec.3.1) by evaluating the throughput (i.e., the number of molecules generated per second). Model throughput was evaluated across a range of batch sizes, with all experiments conducted on a single Nvidia 4090 GPU. As shown in Fig.4, Uni-3DAR consistently outperforms the diffusion-based approach in sampling efficiency, achieving significantly reduced generation times across all tested settings. In particular, at larger batch sizes, Uni-3DAR is approximately 21.8x faster than GeoLDM, and even at a small batch size of 64, it remains about 7.5x faster. Additionally, we assessed the inference overhead introduced by masked next-token prediction. Thanks to our optimizations (Sec. B), we find that masked next-token prediction incurs only a 15% to 30% slowdown. Given its substantial performance gains, this additional cost is well justified.

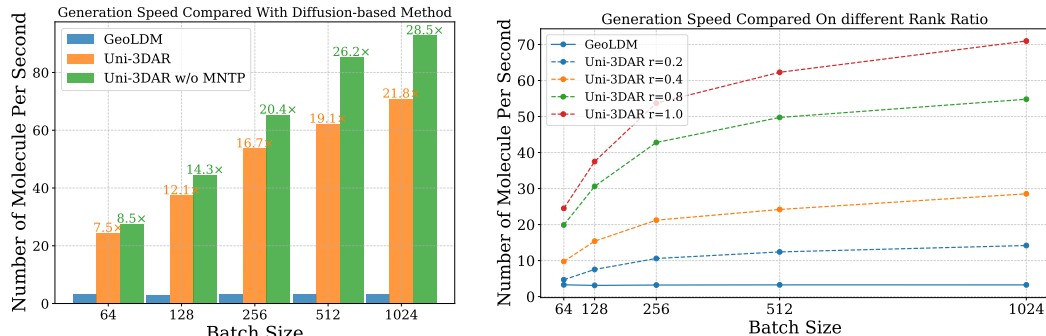

Figure 4: **Left**: Uni-3DAR generation speed on different batch sizes compared with the diffusion-based method; **Right**: Uni-3DAR generation speed on different rank ratios $r$ compared with the diffusion-based method (higher is better).

# E ILLUSTRATION OF THE GENERATED EXAMPLES

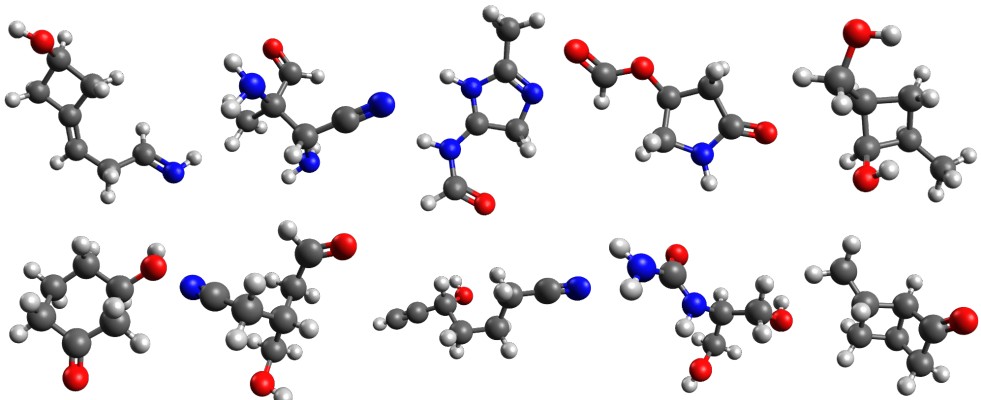

Figure SI-1: Unconditional 3D molecular generation samples of QM9 dataset.

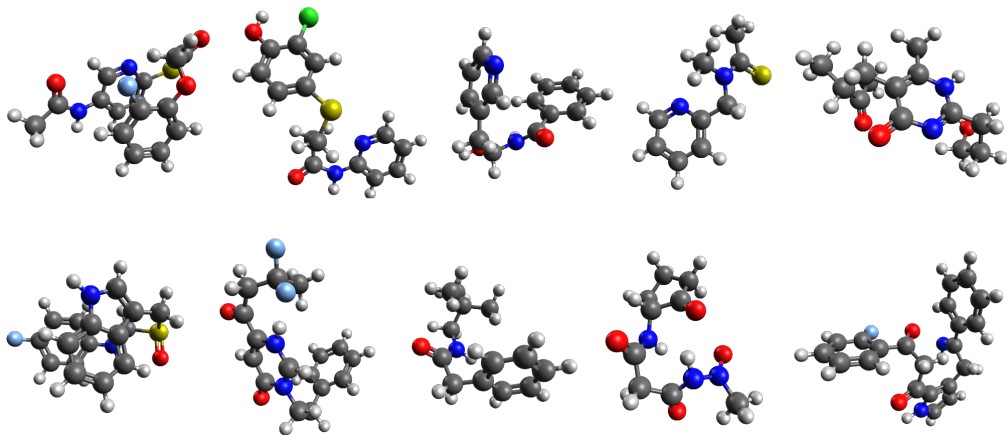

Figure SI-2: Unconditional 3D molecular generation samples of GEOM-DRUG dataset.

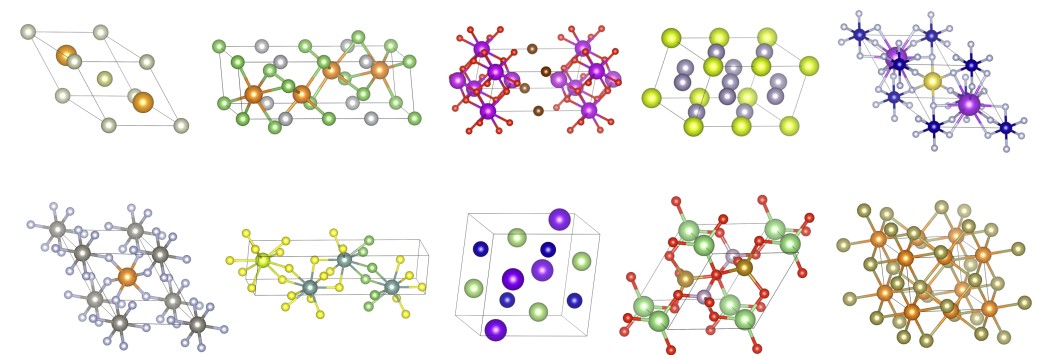

Figure SI-3: De novo crystal generation samples of MP-20 dataset.

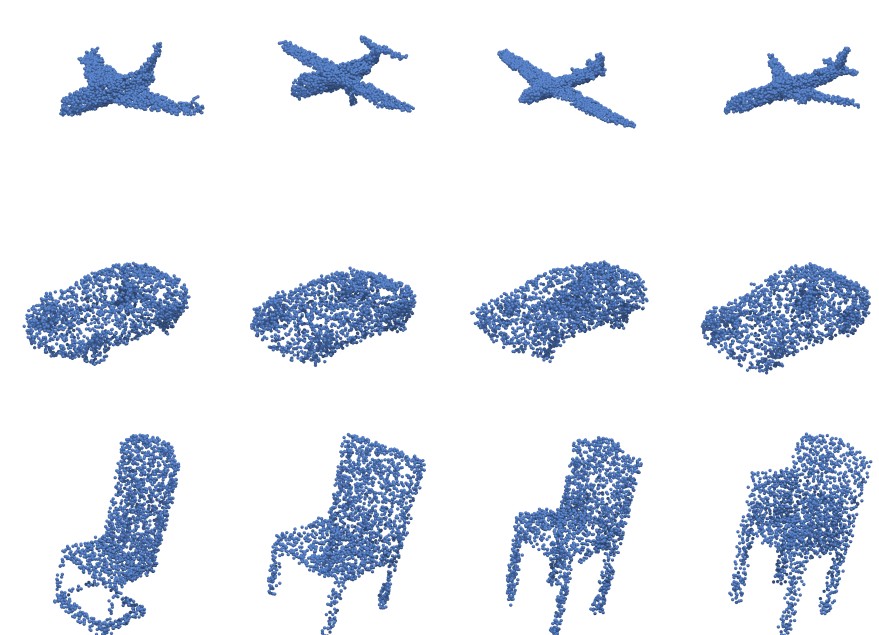

Figure SI-4: Macroscopic 3D object generation samples of ShapeNet dataset.

## F LLM USAGE DETAILS

During the preparation of this manuscript, we utilized Large Language Models (LLMs), Google's Gemini-2.5-pro, for assistance. The use of these models was strictly limited to improving the language and readability of the text. Specific applications included proofreading for grammatical errors, refining sentence structure for clarity, and ensuring a consistent and professional tone throughout the paper. The core scientific ideas, methodologies, experimental results, and conclusions presented in this work were conceived and articulated entirely by the human authors. All AI-generated suggestions were carefully reviewed and edited by the authors to ensure that the final text accurately reflects our original research and intent. The authors take full responsibility for the scientific content and integrity of this paper.

