# OpenReview forum: "Unified Cross-Scale 3D Generation and Understanding via Autoregressive Modeling"
_ICLR.cc/2026/Workshop/FM4Science — ICLR 2026 Workshop FM4Science Poster_

### Official Review · Reviewer_PVrm · 2026-02-23
**Accept: Technically strong cross-scale autoregressive 3D framework with broad empirical coverage**

**Rating:** 7
**Confidence:** 3

**Review:**

# Summary
This paper presents Uni-3DAR, an autoregressive framework for cross-scale 3D generation and understanding. The core method combines (i) coarse-to-fine octree tokenization, (ii) 2-level subtree compression (2LSC), (iii) fine-grained local structural tokens, and (iv) masked next-token prediction (MNTP) to handle dynamic token positions in compressed hierarchical sequences. The authors evaluate the framework across diverse microscopic and macroscopic 3D tasks, spanning generation, conditional reconstruction, and understanding/property prediction (including molecules, crystals, 3D objects, and docking/pocket settings). The paper’s intended contribution is a unified autoregressive interface that can support both generation and understanding tasks across 3D domains.

# Strengths
- Strong ablation evidence for core design choices (Table 12): MNTP improves molecule stability from 88.2 to 93.7, and removing octree tokens causes a major drop (Mol Sta 25.3), supporting the positional and hierarchical tokenization design.
- Performance is competitive and often state-of-the-art on selected metrics across domains, including molecule generation improvements over GeoLDM/UniGEM on key QM9/DRUG metrics (Table 1; e.g., QM9 Mol Sta 93.7, Valid 98.0), crystal CSP gains on MP-20 (Table 3; RMSE 0.0317 vs 0.0566 for FlowMM, with improved match rate), and stronger docking Top-1 results than SurfDock at RMSD <1Å and <2Å (Table 7).
- Efficiency evidence is concrete (Appendix D.3): approximately 7.5x to 21.8x faster generation than GeoLDM on QM9. Although MNTP duplicates tokens, reported overhead is modest after optimization (about 15-30% inference slowdown), and Table 12 shows 6.0h to 6.9h in the matched setting.
- Relative to the paper’s stated closest baseline, the MNTP mechanism is a meaningful contribution: unlike successor-position appending in Octree Transformer, it directly predicts at masked target slots and shows clear gains in ablation.
- 2LSC is practically valuable: deterministic 2-level coding (8-child occupancy pattern -> one 256-way token) yields substantial token-count reduction with a favorable efficiency/performance trade-off.
- The framework is notably broad in scope: it brings a shared autoregressive representation strategy across microscopic and macroscopic 3D settings, and across generation/understanding task families.

# Weaknesses
- The central “unified model” claim is only partially substantiated by current experiments: the paper states separate models are trained per benchmark and joint training is deferred. This is an important gap if unification is interpreted as single-weights multi-task capability.
- Headline claim calibration should be tighter: statements such as broad “state-of-the-art” / “consistently outperforms diffusion-based models” are stronger than what all task-level metrics jointly support.
- Mixed-metric outcomes and limited uncertainty reporting make some conclusions less definitive: pocket prediction on COACH420 is slightly below Vabs-Net (56.2 vs 56.3), ShapeNet gains over LION are small (systematic but narrow), and docking shows a trade-off (better Top-1 but lower Top-5 <2Å than SurfDock, 72.38 vs 75.11); for these close comparisons, mean/std significance reporting is not consistently emphasized.
- Reproducibility is partially addressed but limited by “code upon acceptance,” reducing immediate independent verification.

---

### Official Review · Reviewer_hr7v · 2026-02-24
**Review of Unified Cross-Scale 3D Generation and Understanding via Autoregressive Modeling**

**Rating:** 9
**Confidence:** 4

**Review:**

The paper proposes Uni-3DAR, a unified autoregressive framework for 3D generation and understanding, which generalizes to different domains. With the proposed coarse-to-fine tokenizer, two-level subtree compression strategy, and the masked next-token prediction strategy, Uni-3DAR manifests state-of-the-art performance across multiple 3D generation and understanding tasks with an increased inference speed.

Pros:
- The proposed method generalizes to different domains, providing a unified solution to 3D generation and understanding
- The authors conducted extensive experiments, comprehensively demonstrating the enhanced model performance
- The authors proposed several technical innovations in structural representation, which could be well transferred to other systems

Cons:
- Though generalizable, this method relies on discrete spatial tokenization and lack explicit geometric equivariance, which may limit its physical fidelity for applications such as material structure generation.

---

### Official Review · Reviewer_wnwd · 2026-02-24
**Promising 3D Modeling Framework That May Benefit from Clearer Positioning**

**Rating:** 7
**Confidence:** 3

**Review:**

This paper introduces Uni-3DAR, an autoregressive framework designed to model 3D data ranging from microscopic atomic structures to macroscopic object geometries. The key technical contribution lies in an octree-based tokenization scheme that efficiently exploits spatial sparsity, producing highly compressed representations of 3D structures. Experiments support that this design enables scalable modeling of large molecules and other 3D entities, and offers a potentially useful methodology for scientific tasks that rely on 3D structural information.

**Concerns:**

**1. Motivation and contribution for cross-scale claims**

The notion of unified cross-scale modeling suggests that the framework can either leverage representations learned at one scale to enable fine-tuning at another, or simultaneously handle heterogeneous 3D data and tasks across fundamentally different spatial regimes. However, as stated in the paper's limitations paragraph, true heterogeneous cross-scale modeling remains largely future work.

If the primary strength lies in architectural generality rather than cross-scale transfer or simultaneous modeling, the paper might benefit from reframing the contribution accordingly, e.g., emphasizing scale-agnostic or scale-flexible ability. Clarifying this positioning would make the claims more precisely aligned with the empirical evidence.

**2. Clarity and consistency in Figure 1**

In Figure 1, the level indexing of the octree appears potentially inconsistent with the tree-level convention. It is unclear whether this is typographical or intentional.

---

### Meta-Review · Area_Chair_RLqF · 2026-02-27

**Recommendation:** Accept (Poster)
**Confidence:** 3

**Metareview:**

This paper proposes Uni-3DAR, a unified autoregressive framework for 3D generation and understanding. It seems to be technically strong, with well-motivated hierarchical tokenization, meaningful ablation validation, and solid empirical coverage across multiple 3D tasks, yielding consistent efficiency and accuracy improvements. Reviewers’ main concerns center on calibration of the “cross-scale unified” claim—since models are trained separately per domain—limited uncertainty reporting for close metric comparisons, minor clarity issues, and the absence of explicit geometric equivariance, which may constrain physical fidelity in some scientific settings. Overall, despite some positioning caveats, the work is considered a strong and impactful contribution to scalable autoregressive modeling of sparse 3D data.

---

### Decision · Program_Chairs · 2026-03-03

Accept (Poster)